# The self-interest of adolescents overrules cooperation in social dilemmas

Xiaoyan Wu[1,2,3†], Hongyu Fu[1,2†], Gökhan Aydogan[4], Chunliang Feng[5], Shaozheng Qin[1,2], Yi Zeng[2,6,7,8,9], Chao Liu[1,2]*

[1]State Key Laboratory of Cognitive Neuroscience and Learning, and IDG/McGovernInstitute for Brain Research, Beijing Normal University, Beijing, China; [2]Beijing Key Laboratory of Artificial Intelligence Safety and Superalignment, Beijing, China; [3]Department of Adult Psychiatry and Psychotherapy, University of Zurich, Zurich, Switzerland; [4]Zurich Center for Neuroeconomics, Department of Economics, University of Zurich, Zurich, Switzerland; [5]School of Psychology, South China Normal University, Guangzhou, China; [6]Beijing Institute of AI Safety and Governance, Beijing, China; [7]Brain-inspired Cognitive AI Lab, Institute of Automation, Chinese Academy of Sciences, Beijing, China; [8]University of Chinese Academy of Sciences, Beijing, China; [9]Long-term AI, Beijing, China

*For correspondence:
liuchao@bnu.edu.cn

†These authors contributed equally to this work.

Competing interest: The authors declare that no competing interests exist.

## eLife Assessment

This **important** work investigates cooperative behaviors in adolescents using a repeated Prisoner's Dilemma game. The approach used in the study is **solid**. The impact of this work could be further enhanced with more rigorous modelling procedures and more modeling selection/comparison details, as well as by framing the findings in terms of the specific game-theoretic context, rather than general cooperation. Findings from this study will be of interest to developmental psychologists, economists, and social psychologists.

**Abstract** Cooperation is essential for success in society. Research consistently showed that adolescents are less cooperative than adults, which is often attributed to underdeveloped mentalizing that limits their expectations of others. However, the internal computations underlying this reduced cooperation remain largely unexplored. This study compared cooperation between adolescents and adults using a repeated Prisoner's Dilemma Game. Adolescents cooperated less than adults, particularly after their partner's cooperation. Computational modeling revealed that adults increased their intrinsic reward for reciprocating when their partner continued cooperating, a pattern absent in adolescents. Both computational modeling and self-reported ratings showed that adolescents did not differ from adults in building expectations of their partner's cooperation. Therefore, the reduced cooperation appears driven by a lower intrinsic reward for reciprocity, reflecting a stronger motive to prioritize self-interest, rather than a deficiency in predicting others' cooperation in social learning. These findings provide insights into the developmental trajectory of cooperation from adolescence to adulthood.

## Introduction

Cooperation among individuals facilitates the achievement of shared goals and enhances overall group efficiency (*Fehr and Fischbacher, 2003*; *Nowak, 2006*). For individuals, cooperation skills are key to success in society; this ability is not innate but gradually acquired through socialization

**eLife digest** In everyday life, people often face choices between pursuing their own interests and cooperating with others. Cooperation helps individuals achieve shared goals and build positive relationships, but it often requires sacrificing some immediate personal benefit. Adolescence is a critical period during which young people learn how to manage friendships and collaborate in groups. Many studies have shown that teenagers tend to cooperate less than adults, but the reasons for this remain unclear. Do adolescents struggle to recognise when others are willing to cooperate, or do they recognise these intentions but choose to prioritise their own benefit?

Wu et al. aimed to understand why adolescents cooperate less than adults, both in their behaviour and in the decision-making processes that underlie it. Specifically, they examined whether adolescents fail to recognise cooperative behaviour from others, or whether they do recognise it but are more tempted than adults to take advantage of the situation for personal gain.

To investigate this, the researchers compared the behaviour of teenagers and adults in a repeated cooperation game. In this game, two players could either cooperate for mutual benefit or attempt to gain more for themselves at their partner's expense. The results showed that teenagers cooperated less than adults, particularly after their partner had just cooperated. Importantly, teenagers and adults were equally accurate in estimating how cooperative their partner was. This suggests that adolescents recognise when others are willing to cooperate but feel less motivated to reciprocate.

These findings may help teachers, parents, and those designing school programmes better support teenagers' social development. The study of Wu et al. suggests that it may be useful not only to help adolescents understand others' intentions, but also to strengthen the value they place on fairness and on reciprocating cooperation when others behave kindly. Future research should explore whether similar patterns occur in real-life interactions and across more diverse groups of young people.

(*Warneken, 2018*). Successful cooperation requires individuals to prioritize the common purpose over their personal interests, focusing on collective goals (*Sachs et al., 2004*). Experimental psychology has often used the Prisoner's Dilemma Game (PDG; *Axelrod and Hamilton, 1981*) to study human cooperative behaviors. Extensive research has adapted the PDG into a repeated version to explore how people respond to interactive cooperation (*Andreoni and Miller, 1993*; *Embrey et al., 2018*), requiring individuals to adjust their responses dynamically to others and simulating real-life cooperation more closely (*Axelrod and Hamilton, 1981*). In such social dilemmas, individuals face a trade-off between immediate rewards from defection and long-term benefits from cooperation (*Rilling et al., 2002*). Decision-making in these situations is thought to engage mentalizing abilities, which are functions related to theory of mind that enable individuals to form expectations about others' cooperative intentions (*Rilling et al., 2004*).

Cooperation is not an innate skill but is gradually cultivated and refined through socialization (*House et al., 2020*). Adolescence, in particular, marks a critical developmental phase in the transition to independent social roles (*Steinberg, 2005*). Studies using the PDG consistently show that adolescents cooperate less than adults (*Belli et al., 2012*; *Nava et al., 2023*; *Taheri et al., 2018*). This reduced cooperation is often attributed to an underdeveloped theory of mind, which may lead adolescents to underestimate others' trustworthiness and willingness to cooperate in social learning (*Gutiérrez-Roig et al., 2014*; *Fett et al., 2014*).

However, there are findings that may not support this hypothesis. For example, a previous study found that adolescents' lower cooperation, compared to adults, emerges only when following a partner's cooperation. Conversely, when the partner defected, adolescents' cooperative behaviors resembled those of adults (*Gutiérrez-Roig et al., 2014*). Similarly, a Trust Game study (*Fett et al., 2014*) reported a comparable pattern: adolescents invested less (measured as trust behavior) than adults only when the partner was cooperative. When faced with a non-cooperative partner, both adolescents and adults consistently reduced their trust behaviors. These findings suggest that adolescents, like adults, are able to adjust their behavior in response to others' actions. This selective reduction in adolescent cooperation implies that factors beyond deficits in mentalizing may be at play. Adolescents may prioritize maximizing immediate rewards over long-term reciprocity (*Rilling et al., 2002*). When confident that their partner will cooperate, defection may become the optimal strategy

for maximizing self-interest. This hypothesis remains untested, but computational modeling could provide a valuable approach for examining the underlying mental processes behind these behavioral variations (*Farrell and Lewandowsky, 2010*; *Wu et al., 2024a*; *Wu et al., 2024b*).

This study aimed to investigate variations in cooperative behavior between adolescents and adults and to explore the mental processes underlying these differences using computational modeling. Based on legal criteria for majority and prior empirical work, we adopt 18 years as the boundary between adolescence and adulthood (*Icenogle et al., 2019*; *Tervo-Clemmens et al., 2023*). A total of 127 adolescents and 134 adults participated in the study, playing a repeated Prisoner's Dilemma Game (rPDG) with a presumed human partner, whose behavior was predetermined by a computer program (see *Figure 1a*). The program ensured consistent conditions across age groups. To enhance realism, variability was introduced into the computer-simulated partner's behavior. The rPDG provides a symmetric and simultaneous framework that isolates the motivational conflict between self-interest and joint welfare, avoiding the sequential trust and reputation dynamics characteristic of asymmetric tasks such as the Trust Game (*Rilling et al., 2002*; *King-Casas et al., 2005*). Based on the standard payoff matrix of the rPDG (*Figure 1b*), mutual cooperation maximizes collective interests, while defection maximizes self-interest from an individual perspective. Our focus was on how adolescents respond to their partner's consistent cooperation and defection, aiming to identify potential mental variables contributing to adolescents' lower cooperation.

We developed computational models to investigate the dynamic variables guiding cooperative decisions in the rPDG. The model explicitly incorporates both expectations of the partner's cooperation and the intrinsic reward of reciprocity. A basic reinforcement learning (RL) algorithm was used to model participants' dynamic expectations regarding the partner's cooperation. Drawing on research on asymmetric reward learning in adolescents (*Palminteri et al., 2016*; *Rosenbaum et al., 2022*), we included asymmetric updating for positive (better-than-expected) and negative (worse-than-expected) outcomes. We identified the asymmetric RL learning model as the winning model that best explained the cooperative decisions of both adolescents and adults. Participants' expectations were modeled as a trial-by-trial dynamic variable, represented by parameter $p$. Following previous studies (*Fareri et al., 2012*; *Fareri et al., 2015*), a non-monetary reward for cooperation, represented by parameter $\omega$, reflects individual preferences for mutual cooperation. The term $p \times \omega$ quantifies the intrinsic reward for reciprocity.

We hypothesize that adolescents will exhibit lower overall cooperation compared to adults, consistent with previous studies (*Belli et al., 2012*; *Nava et al., 2023*; *Taheri et al., 2018*). Specifically, we expect adolescents to demonstrate reduced cooperation after their partner's cooperation but not following defection (*Gutiérrez-Roig et al., 2014*; *Fett et al., 2014*). Furthermore, we aim to explore whether this lower conditional cooperation is driven by inappropriate expectations of their partners (represented by $p$), a reduced intrinsic reward for reciprocity (represented by $p \times \omega$), or a combination of both.

## Results

### Adolescents exhibit lower cooperation than adults following partner cooperation, but not defection

In each trial of the rPDG, as shown in *Figure 1c*, participants were presented with two choices: a triangle representing cooperation and a square representing defection. The choice associated with each symbol was randomly balanced across participants. They were informed that they were playing the game simultaneously with another partner. After making their decision, participants were shown both their own choice and that of their partner. We performed a generalized linear mixed model (GLMM1) analysis (see *Appendix 1—table 1*) to examine the effects of each independent variable and their interactions on the decision to cooperate or defect.

Consistent with most previous studies (*Belli et al., 2012*; *Nava et al., 2023*; *Taheri et al., 2018*), adolescents cooperated less than adults ($b$ of group = 0.79, 95% CI = [0.311, 1.270], p = 0.001; *Figure 1d*). Following the interaction of group × previous trial × partner's choice ($b$ of interaction = 0.24, 95% CI = [0.126, 0.361], p < 0.001), we found that adolescents showed significantly less cooperation compared to adults only after the partner's cooperation ($t(259)_{group}$ = −2.84, p = 0.005, BF10=6.01).

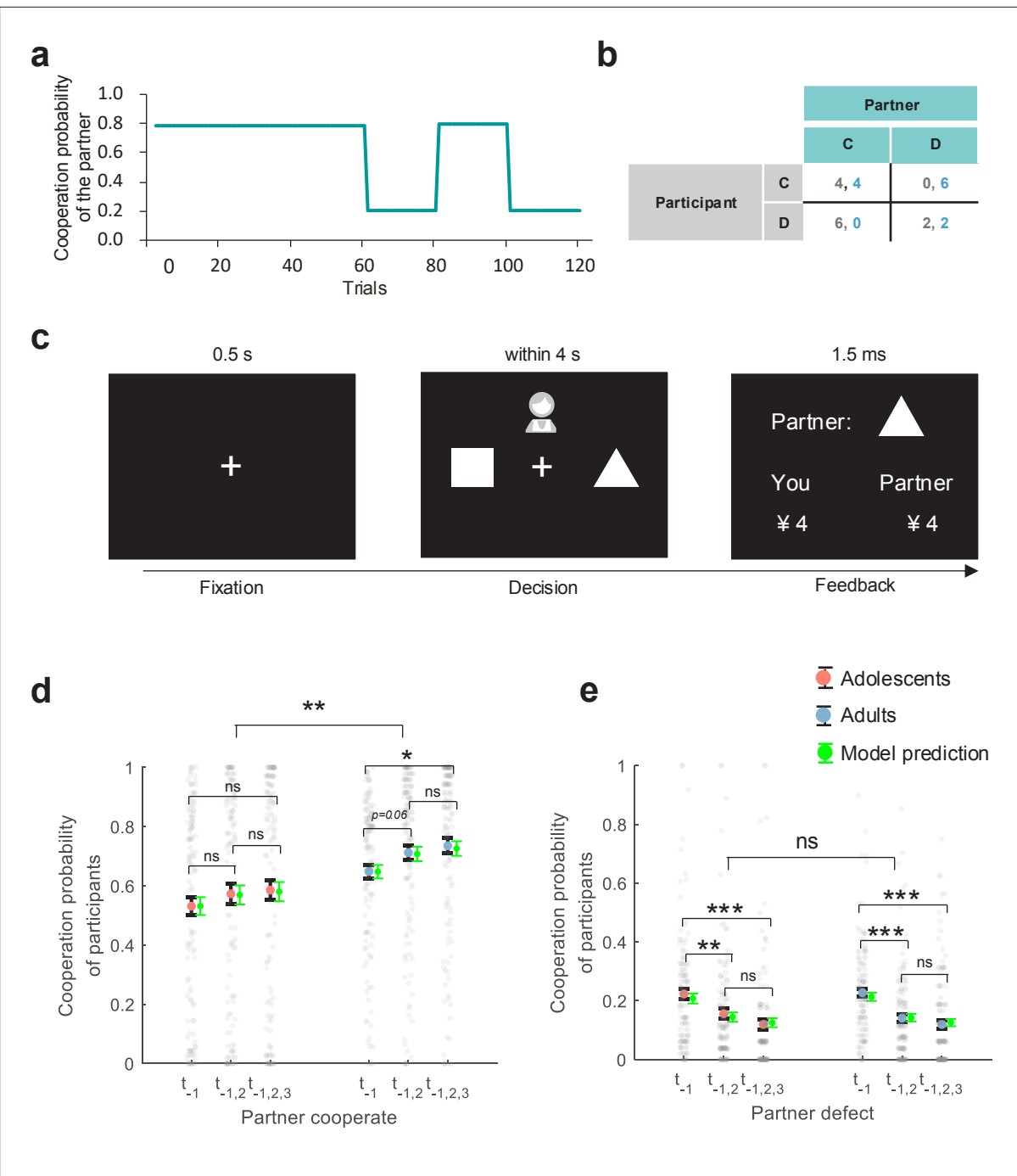

**Figure 1.** Experiment setup and behavioral results. (**a**) Partner's cooperation probability: in the first half of the 120 trials, the partner cooperated 78% of the time; in the second half, cooperation alternated between 20% and 80%. (**b**) Payoff matrix: payoffs are 4 for mutual cooperation, 2 for mutual defection, 0 for cooperation when the other defects, and 6 for defecting when the other cooperates. (**c**) Trial illustration: after a 0.5 s fixation, participants choose a shape (triangle for cooperation, square for defection) within 4 s and see both players' choices for 1.5 s. (**d, e**) Post hoc comparisons: **d** and **e** show the participants' cooperation probability on the y-axis. The x-axis represents the consistency of the partner's actions in previous trials ($t_{-1}$: last trial, $t_{-1,2}$: last two trials, $t_{-1,2,3}$: last three trials). Large red (adolescents) and blue (adults) dots indicate mean probabilities, with black error bars for standard error (SE). Gray dots represent mean probabilities across trials, and green error bars show predicted cooperation rates with SE. Notes: *n.s.* p>0.05; *p<0.05; **p<0.01; ***p<0.001.

However, such a difference was not significant after the partner's defection ($t(259)_{group} = -1.86$, $p = 0.064$, BF10=0.69; *Figure 1d*). We also found that adults increased cooperation in response to their partners' consistent cooperation (the partner cooperated once vs. the partner cooperated thrice: $t(266)_{adults} = -2.50$, p = 0.013, BF10=2.56), but this pattern was not observed in adolescents ($t(252)_{adolescents} = -1.18$, p = 0.239, BF10=0.27, see *Figure 1d*).

Nevertheless, both groups significantly decreased cooperation in response to the partner's continual defection (the partner defected once vs. the partner defected twice: $t(266)_{adults} = 4.46$, p < 0.001, BF10 >$10^3$, $t(252)_{adolescents} = 2.78$, p = 0.006, BF10=5.21; the partner defected once vs. the partner defected thrice: $t(266)_{adults} = 5.56$, p < 0.001, BF10 >$10^3$ for adults, $t(252)_{adolescents} = 4.32$, p < 0.001, BF10=761.12 for adolescents, *Figure 1e*).

## Asymmetric RL learning in the social reward model best explains cooperative decisions of adolescents and adults

Computational modeling was used to simulate participants' mental processes during the rPDG. Starting with a baseline model that assumed decisions were made through random selection (Model

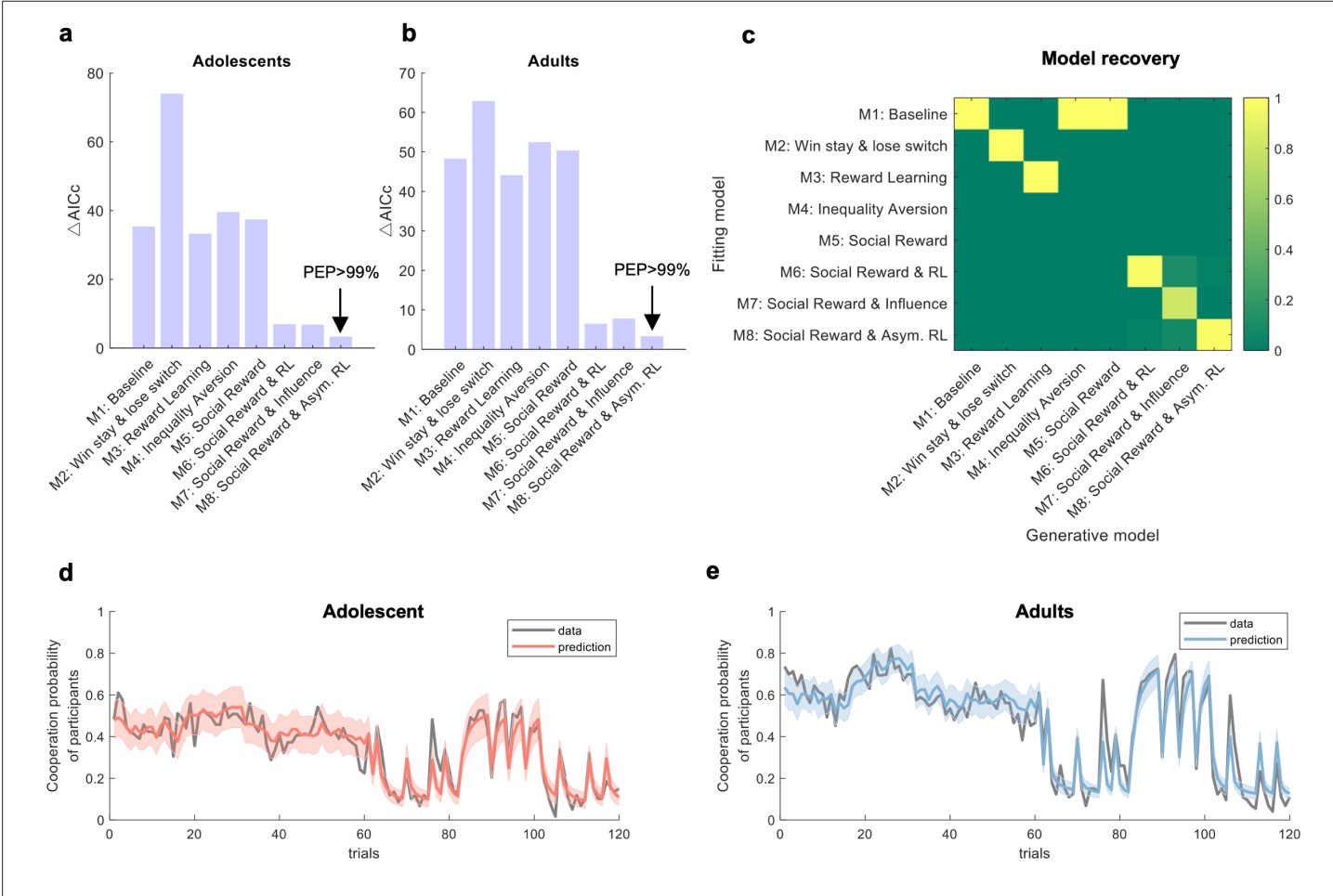

**Figure 2.** Computational modeling. (**a, b**) Model comparisons for adolescents and adults, respectively. The y-axis represents model fitness based on the Akaike Information Criterion with a correction for sample size (AICc; *Hurvich and Tsai, 1989*). For each participant, the model with the lowest AICc served as a reference to compute ΔAICc by subtracting it from the AICc of other models ($\Delta \mathbf{AICc} = \mathbf{AICc}_x - \mathbf{AICc}_{lowest}$). A lower ΔAICc indicates a better model fit. Protected exceedance probability (PEP) is a group-level measure that assesses the likelihood of each model's superiority over the others (*Rigoux et al., 2014*). (**c**) Model recovery analysis. Each model was used to generate 100 synthetic datasets, and for each dataset, model fitting and comparison were performed. Each column corresponds to one generative model, and each row corresponds to one fitting model. The color in each cell indicates the probability that the synthetic datasets generated by the model in the column were best fit by the model in the row, with a darker color denoting a higher probability. (**d, e**) Model prediction. Sample illustration of the best-fitting model prediction versus data for adolescents and adults, respectively.

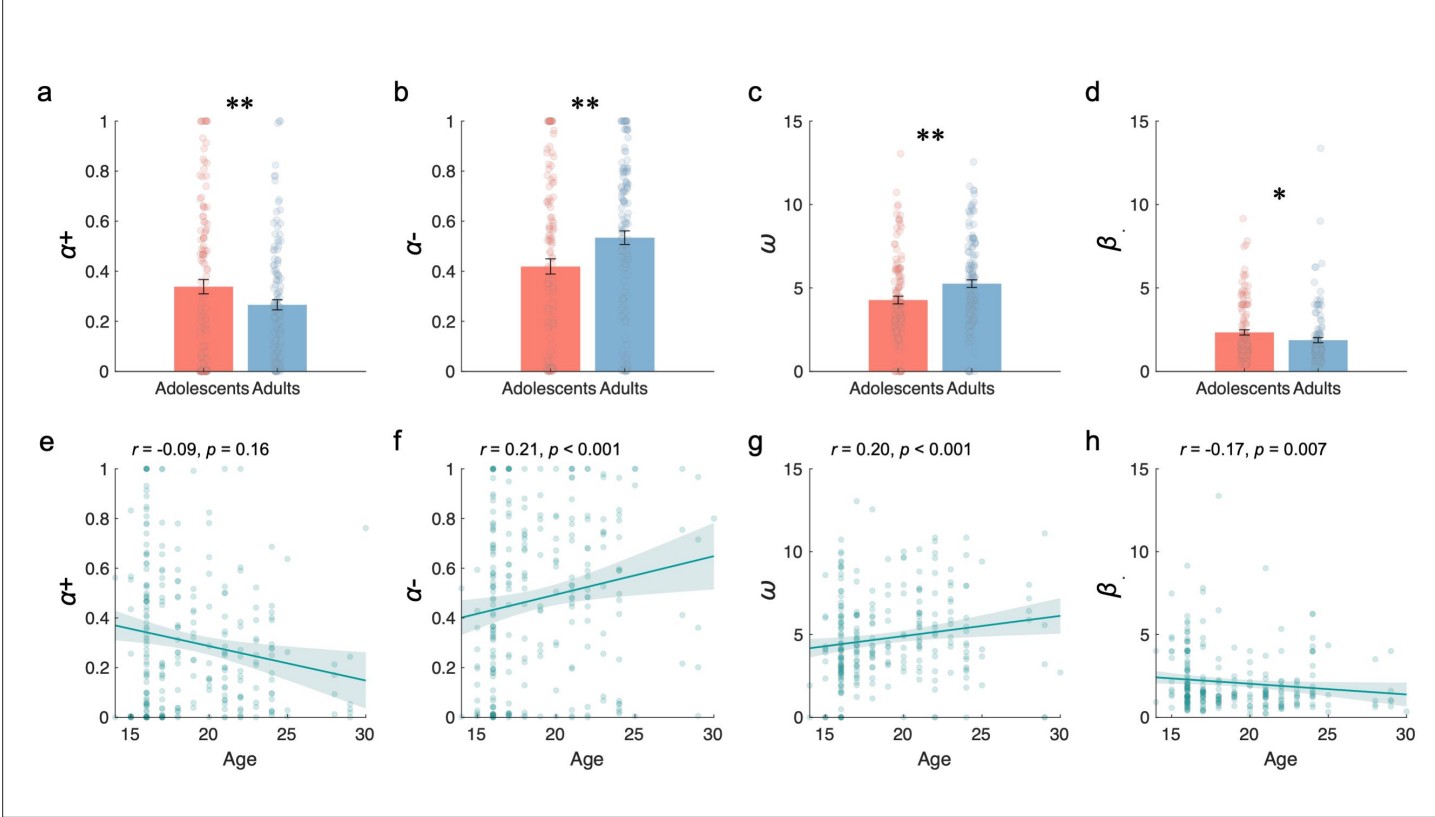

**Figure 3.** Learning rates and social preferences. (**a–d**) Comparison between adolescents and adults for positive learning rate ($\alpha+$), negative learning rate ($\alpha-$), social preference ($\omega$), and inverse temperature ($\beta$), respectively. (**e–h**) Correlation between age and positive learning rate, negative learning rate, social preference, and inverse temperature, respectively. Notes: *p<0.05; **p<0.01.

1), we compared several alternatives: a win-stay and loss-shift model (Model 2), a reward learning model (Model 3), an inequality aversion model (Model 4), and a social reward model (Model 5). Among these, the social reward model outperformed the others. We then compared a basic RL algorithm (Model 6), an influence learning rule (Model 7), and an asymmetric RL learning rule (Model 8) within the social reward framework. The asymmetric RL learning model best explained the cooperative decisions of both adolescents and adults (see *Figure 2a* for adolescents and *Figure 2b* for adults; methods for details). Model recovery analysis indicated that the asymmetric RL learning within the social reward model was distinguishable from the other models (*Figure 2c*) and accurately captured the behaviors of both adolescents (*Figure 2d*) and adults (*Figure 2e*). The overlap between Models 4 and 5 likely arises because neither model incorporates a learning mechanism, making them less able to account for trial-by-trial adjustments in this dynamic task. For further validation of the best-fitting model, see *Appendix 1—figure 1* for model predictions, *Appendix 1—figure 2* for the distributions of free parameters, *Appendix 1—figure 3* for parameter recovery, *Appendix 1—figure 4* for partial correlation matrices among parameters, and *Appendix 1—figure 5* for group-level posterior distributions from the hierarchical Bayesian estimation for the best-fitting model.

## Distinct learning rates and social preferences between adolescents and adults in repeated cooperation

Although the asymmetric RL learning in the social reward model best explained the behaviors of both adolescents and adults, the two groups exhibited distinct learning dynamics and social preferences for cooperation. Specifically, adolescents applied a higher positive learning rate ($\alpha+$, $t(259) = 2.95$, p = 0.003, BF10=8.02, *Figure 3a*) to update better-than-expected prediction errors, and a lower negative learning rate ($\alpha-$, $t(259) = -2.62$, p = 0.009, BF10=3.46, *Figure 3b*) for worse-than-expected prediction errors.

Additionally, a positive correlation was found between participants' age and the negative learning rate ($\alpha-$, $r = 0.21$, p < 0.001, *Figure 3f*), while no significant correlation was observed with the positive learning rate ($\alpha+$, $r = -0.09$, p = 0.16, *Figure 3e*).

Furthermore, adolescents displayed a weaker preference for cooperation compared to adults ($\omega$, $t(259) = -3.03$, p = 0.003, BF10=9.92, *Figure 3c*), and their social preferences for cooperation increased with age ($r = 0.20$, p < 0.001, *Figure 3g*). Additionally, adolescents exhibited a higher inverse temperature parameter compared to adults, indicating they were more sensitive to utility differences between cooperation and defection ($\beta$, $t(259) = 2.14$, p = 0.034, BF10=1.17, *Figure 3d*). This sensitivity decreased with age, as shown by a negative correlation with age ($r = -0.17$, p = 0.007, *Figure 3h*).

## Adolescents compared to adults show no inappropriate expectations but less intrinsic reward for reciprocity

To further explore what underlies the observed decrease in cooperation among adolescents, we focused on two hidden trial-by-trial updating variables: the partner cooperation expectation ($p$) and the intrinsic reward for reciprocity ($p \times \omega$). Additionally, participants' self-reported cooperativeness scores, assessed every 15 trials, provided further insight into their subjective estimation of the partner's willingness to cooperate.

### Partner cooperation expectation

We performed a linear mixed model (LMM1, *Appendix 1—table 2*) on partner cooperation expectation to assess the effects of each independent variable and their interactions. Following the interaction of group × previous trial × partner's choice ($b$ of interaction = 0.03, 95% CI = [0.022, 0.038], p < 0.001), we found that the partner cooperation expectation for both adolescents and adults increased with the partner's consistent cooperation (the partner cooperated once vs. the partner cooperated twice: $t(252)_{adolescents} = -2.81$, p = 0.005, BF10=5.75, $t(266)_{adults} = -4.45$, p < 0.001, BF10 >$10^3$; the partner cooperated once vs. the partner cooperated thrice: $t(252)_{adolescents} = -3.69$, p < 0.001, BF10=78.00, $t(266)_{adults} = -6.23$, p < 0.001, BF10 >$10^3$; *Figure 4a*). Additionally, expectations decreased with the partner's consistent defection (the partner cooperated once vs. the partner cooperated twice: $t(252)_{adolescents} = 4.44$, p < 0.001, BF10 >$10^3$, $t(266)_{adults} = 7.02$, p < 0.001, BF10 >$10^3$; the partner cooperated once vs. the partner cooperated thrice: $t(252)_{adolescents} = 5.60$, p < 0.001, BF10 >$10^3$, $t(266)_{adults} = 8.40$, p < 0.001, BF10 >$10^3$; *Figure 4b*). These results showed that both adolescents and adults held very similar expectations toward their partner's cooperation and did not have significant differences between the groups ($b$ of group = –0.04, 95% CI = [–0.102, 0.021], p = 0.198).

Moreover, we performed an LMM2 (*Appendix 1—table 3*) analysis on participants' self-reported scores regarding the cooperativeness of their partners to examine the effects of each independent variable and their interactions. In line with the expectation of partner cooperation, we observed minimal discrepancy in the self-reported scores on partner cooperativeness between adolescents and adults. Neither the main effect of group nor the interaction achieved statistical significance ($b$ of group = 0.17, 95% CI = [–0.51, 0.85], p = 0.616; $b$ of interaction = 0.38, 95% CI = [–0.052, 0.812], p = 0.085; *Figure 4c–d*). These results provide evidence that adolescents did not differ from adults in assessing their partner's cooperation.

### Intrinsic reward for reciprocity

We performed an LMM3 (*Appendix 1—table 4*) on the intrinsic reward for reciprocity to assess the effects of each independent variable and their interactions. We found that adolescents appreciated reciprocity less than adults did ($b$ of group = 0.52, 95% CI = [0.224, 0.816], p < 0.001).

Following the interaction of group × previous trial × partner's choice ($b$ of interaction = 0.37, 95% CI = [0.318, 0.424], p < 0.001), unlike adults, adolescents did not increase their intrinsic reward for reciprocity in response to the partner's consistent cooperation (the partner cooperated once vs. the partner cooperated twice: $t(252)_{adolescents} = -0.96$, p = 0.336, BF10=0.21, $t(266)_{adults} = -2.13$, p = 0.034, BF10=1.15; the partner cooperated once vs. the partner cooperated thrice: $t(252)_{adolescents} = -1.38$, p = 0.170, BF10=0.34, $t(266)_{adults} = -3.08$, p = 0.002, BF10=11.63; *Figure 4e*).

Moreover, under consistent defection by the partner, evidence for the one-versus-two last trials comparison was inconclusive in adolescents but supported a decrease in adults ($t(252)_{adolescents} = 1.99$,

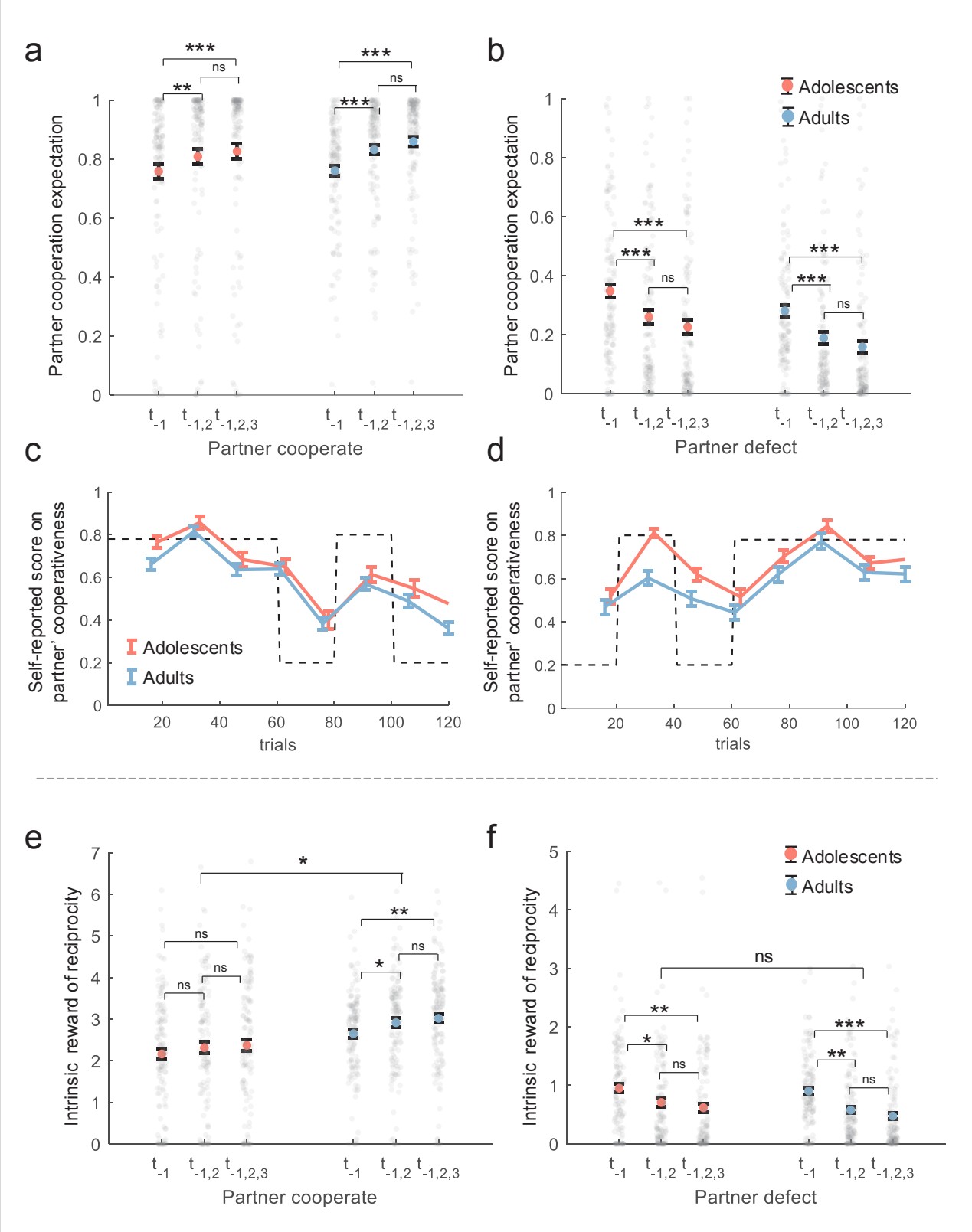

**Figure 4.** Analysis of hidden variables from the best-fitting model. (**a, b**) Post-hoc comparison of LMM1: interaction of group × previous trial × partner's choice. The y-axis shows participants' expectations of partner cooperation probability (*p*) from the best-fitting model. (**c, d**) Self-reported cooperativeness: normalized scores on partner cooperativeness for two orders of partner cooperation probability, with adolescents (orange-red line) and adults (blue line). Scores were assessed on a 0–9 scale and normalized to 0–1. The dotted line indicates the presumed partner's cooperation

*Figure 4 continued on next page*

Figure 4 continued

probability, with mean values and standard errors shown. (**e, f**) Post-hoc comparison of LMM3: interaction of group × previous trial × partner's choice. The y-axis shows participants' intrinsic reward for reciprocity ($p \times \omega$) from the best-fitting model. The x-axis represents the consistency of the partner's actions in previous trials ($t_{-1}$: last trial, $t_{-1,2}$: last two trials, $t_{-1,2,3}$: last three trials). Colored dots with error bars indicate mean values with standard errors for adolescents (orange-red) and adults (blue), while small gray dots represent individual participants. Notes: *n.s.*p>0.05; *p<0.05; **p<0.01; ***p<0.001.

p = 0.047, BF10=0.90, $t(266)_{\text{adults}} = -2.71$, p = 0.007, BF10=4.27). Importantly, in the one-versus-three last trials comparison, both adolescents and adults consistently showed a decrease in intrinsic reward for reciprocity ($t(252)_{\text{adolescents}} = 2.64$, p = 0.009, BF10=3.66, $t(266)_{\text{adults}} = 3.37$, p < 0.001, BF10=27.47; *Figure 4f*).

In brief, adolescents did not deviate in forming expectations about their partner's willingness to cooperate, but they showed lower social preferences for cooperation and a reduced intrinsic reward for reciprocity. Specifically, compared to adults, adolescents displayed less intrinsic reward for reciprocity and did not increase it in response to consistent cooperation, although their reactions to consistent defection tended to be similar to those of adults.

## Discussion

Cooperation lies at the heart of societal functioning, facilitating the achievement of shared goals and fostering social harmony. In this study, we sought to deepen our understanding of the developmental aspects of cooperation by examining differences in cooperative behavior between adolescents and adults in the context of the rPDG. Our findings shed light on the cognitive and affective processes underlying these behaviors, offering insights into the mechanisms driving cooperative decision-making across different developmental stages.

Consistent with many previous studies (*Fett et al., 2014*; *Gutiérrez-Roig et al., 2014*; *Westhoff et al., 2020*), our results showed that adolescents exhibited lower levels of cooperation compared to adults. However, such lower cooperation was not generally observed during the task, but selectively occurred after their partner cooperated in the previous rounds. Moreover, our results showed that adults increased cooperation in response to their partner's consistent cooperation; such a pattern was not observed in adolescents. However, both age groups decreased cooperation in response to consistent partner defection, indicating shared responses to non-cooperative behavior.

Our results suggest that the lower levels of cooperation observed in adolescents stem from a stronger motive to prioritize self-interest rather than a deficiency in predicting others' cooperation in social learning. In both, the expectation of partner's cooperation estimated from computational modeling and the self-reported measurements, adolescents did not exhibit significant differences from adults. However, adolescents exhibited a weaker preference for (conditional) cooperation compared to adults, resulting in a reduced intrinsic reward for reciprocity. The results are consistent with prior research (*Crone and Dahl, 2012*; *Do et al., 2017*; *Pfeifer and Berkman, 2018*; *van den Bos et al., 2010*, *van den Bos et al., 2011*), suggesting that adolescents prioritize immediate gains over long-term benefits, potentially undermining the benefits of cooperation. This tendency aligns with earlier findings that adolescents exhibit heightened sensitivity to reward feedback (*Blakemore and Mills, 2014*; *Crone and Dahl, 2012*; *Davis et al., 2023*; *Do et al., 2017*; *van den Bos et al., 2011*; *van Duijvenvoorde et al., 2015*), which may influence their decision-making in cooperative interactions. Overall, these findings indicate that adolescents' lower cooperation is unlikely to be driven solely by strategic considerations, but may instead reflect differences in the valuation of others' cooperation or reduced motivation to reciprocate. Although defection is the payoff-dominant strategy in the PDG, the selective pattern of adolescents' cooperation and the model comparison results indicate that their reduced cooperation cannot be fully explained by strategic incentives, but rather reflects weaker valuation of social reciprocity.

It has been acknowledged that individuals update positive and negative outcomes by different weights in social cooperation, and such asymmetric learning process can be modeled by a basic RL algorithm with both positive and negative learning rates (*Garrett and Daw, 2020*; *Rosenbaum et al., 2022*). In this study, we find that an asymmetrical RL algorithm in a social reward model provided best model fits of the behaviors of both adolescents and adults. Adolescents demonstrated a larger positive learning rate, but a smaller negative learning rate compared to adults, suggesting heightened

sensitivity to positive feedback from cooperative behavior and reduced sensitivity to negative feedback from defection. This asymmetrical learning pattern may drive adolescents to focus more on self-beneficial social signals, maximizing immediate gains in response to cooperative behavior. These findings align with (*van den Bos et al., 2011*), which highlight adolescents' heightened sensitivity to immediate rewards and less stable trusting behavior compared to adults. Adolescents also showed higher inverse temperature values ($\beta$), indicating greater sensitivity to expected value and more value-based choice behavior. Together, these findings suggest that the differentiation between positive and negative learning rates changes with age, reflecting more selective feedback sensitivity in development, while higher ($\beta$) values in adolescents indicate greater value sensitivity. This interpretation remains tentative and requires further validation in future research.

Adolescence is characterized by increased self-discovery and egocentrism (*Pfeifer and Berkman, 2018*; *Ting et al., 2019*), leading individuals to prioritize immediate gains over long-term benefits. Consistent with this, the higher value sensitivity ($\beta$) observed in adolescents suggests a stronger focus on immediate utility during cooperative exchanges. Consequently, adolescents may be more inclined toward self-serving motives in sustained social interactions (*Pfeifer and Berkman, 2018*). However, these tendencies are not static; as individuals mature into adulthood, their socio-emotional capacities continue to develop (*Worthman and Trang, 2018*), enabling a more balanced integration of short-term rewards and long-term social outcomes (*Crone and Dahl, 2012*; *Wu et al., 2023*).

It is important to note some limitations of this study. First, we used artificial opponents with predetermined cooperation patterns to better control the stimuli. While this approach allowed us to isolate specific motivations for cooperation (financial vs. social rewards), it is possible that participants might behave differently in more natural settings. Our study serves as an initial step in understanding cooperation motivations in adolescents and adults, and future research could explore these behaviors in more real-world contexts. Second, our study employed the rPDG as the primary task to directly capture cooperation in symmetric multi-round interactions. However, because it is a zero-sum framework that structurally incentivizes defection as the dominant strategy, the rPDG may influence choices beyond participants' intrinsic preferences. For example, one potential interpretation of adolescents' lower cooperation is that they adopt a strategic response to the payoff structure, through leveraging defection as the more rewarding strategy within the game. If this account holds, adolescents should exhibit lower cooperation across all rounds. However, we find that adolescents and adults exhibit similar behavioral patterns when partners defect. By contrast, adolescents cooperate less than adults when partners cooperate, and their cooperation does not increase significantly even when partners cooperate consecutively. Although this pattern is consistent with the interpretation that adolescents' lower cooperation reflects a relatively more self-interested motivation, stronger conclusions about age differences in cooperative preferences require further examination in tasks with varied structures. Third, although both age groups were recruited from Beijing and nearby regions, minimizing major regional and cultural variation, adolescents and adults may still differ in socioeconomic status, financial independence, and social experience. Such contextual differences could interact with developmental processes in shaping cooperative behavior and reward valuation. Future research with demographically matched samples or explicit measures of socioeconomic background will help disentangle biological from sociocultural influences.

In conclusion, our study contributes to an understanding of the developmental aspects of cooperation and the cognitive-affective processes underlying cooperative decision-making. By examining differences in cooperative behavior between adolescents and adults in the rPDG and integrating computational modeling, we offer valuable insights into the mechanisms driving cooperative behavior across different developmental stages. These findings have implications for promoting prosocial behaviors and designing effective socialization interventions during adolescence. By highlighting the importance of reciprocity, our findings offer insights into the developmental trajectory of cooperation from adolescence to adulthood and provide practical implications for enhancing cooperative interactions in real-world contexts.

## Materials and methods

### Participants

A total of 261 participants took part in the current study, consisting of 127 adolescents (n=127, aged 14–17 years, mean ± SD: 16.13±0.63, 44 females) and 134 adults (n=134, aged 18–30 years, mean ± SD: 21.63±2.88, 79 females). No a priori power analysis was conducted. The sample size was determined based on previous studies investigating cooperation behaviors in adolescents and adults. Adolescents were recruited from a local high school, and adults were recruited through advertisements on a university campus forum. Written informed consent was obtained from all adult participants. For adolescents, written informed consent was obtained from their legal guardians, and assent was obtained from the adolescents themselves prior to participation. Participants were included if they had normal or corrected-to-normal vision and no history of psychiatric or neurological illness. Exclusion criteria included any self-reported diagnosed psychiatric or neurological disorder. No participants dropped out of the experiment, and all collected data were included in the statistical analyses. This study was approved by the Ethics Committee of Beijing Normal University (Approval Nos. CNL_A_0001_009 and RB_A_0003_202001). All procedures were conducted in accordance with the Declaration of Helsinki. Participants received monetary compensation based on their task performance (see rPDG for details).

### Experimental procedure

All participants completed the experiments in a laboratory setting with multiple participants present. They were informed that they were participating in a multiple-round interaction game with an anonymous partner. In the instructions section, we referred to the interaction game as the rPDG and refrained from using the terms 'cooperate' and 'defect' to minimize the influence of social expectations, biases, and promote comparability across studies. Participants were instructed to believe that their partner was also playing the game at the same time. Compensation for their participation was based on the tokens earned during the game, with 10% of the rounds randomly selected for payment calculation at an exchange rate of 1 token to 1 yuan. Participants were explicitly informed in advance about this incentive mechanism. Prior to the formal experiment, participants underwent a quiz and several practice rounds to ensure a full understanding of the task. Following the experiment, participants completed a Social Value Orientation (SVO) task to assess their prosocial personality traits. The entire procedure lasted approximately 60 min. Blinding was not applicable in this study, as all participants interacted with a computer-controlled partner. To minimize potential bias, the partner's behavior patterns and stimulus meanings were randomized across participants. A detailed protocol is available upon request.

### The repeated prisoner's dilemma game

Similarly to the classic version of PDG, rPDG involves two players. Consistent with the standard payoff matrix of the PDG (*Figure 1b*), when both players cooperated (defected), they each received four tokens (two tokens). If the players made different decisions, the one who cooperated received 0 tokens, while the one who defected received 6 tokens. Participants were told that their partner was another human participant in the laboratory and that they would interact with the same partner across all rounds. However, in reality, the actions of the partner were predetermined by a computer program. This setup allowed for a clear comparison of the behavioral responses between adolescents and adults. Participants were not informed of the total number of rounds in the rPDG. In order to enhance the realism of the partner's response, we manipulated the variability in the partner's decision making. The partner's cooperation probability remained stable at 78% for half of the trials. In the other half of the trials, the partner's cooperation probability varied, switching between 20%, 80%, and 20% for each set of 20 trials. The order of these two sessions was counterbalanced between participants. During the rPDG, participants were asked every 15 rounds to evaluate their partner's cooperativeness using a 10-point scale, where 0 represents 'no cooperation' and 9 represents 'very high cooperation'. The question posed to the participants was 'How cooperative do you think your partner is at the moment?'.

### Behavioral data analysis

All statistical analyses were conducted in MATLAB R2023a (RRID:SCR_001622). GLMM was implemented using the 'fitglme' function in MATLAB. Interaction contrasts were performed for significant

interactions and, when higher-order interactions were not significant, pairwise or sequential contrasts were performed for significant main effects. Post hoc comparisons were conducted using Bayes factor analyses with MATLAB's bayesFactor Toolbox version v3.0, with a Cauchy prior scale $\sigma = 0.707$ (**Krekelberg, 2024**).

GLMM1: Participant's choices (cooperate or defect) of all trials are the dependent variable; fixed effects include an intercept, the main effects of group (adolescents or adults), previous trial (last one trial, last two trials, and last three trials), partner's choice (cooperation or defection), and all possible interaction effects of the independent variables. Gender (male and female) and timing (trial number from 1 to 120) were also included as the control variables. Random effects include correlated random slopes of group, previous trial, partner's choice, gender, trial number, and random intercept for participants. The group, previous trial, partner's choice, and gender are the category variables. The trial number is a continuous variable. See **Appendix 1—table 1** for the statistical results of GLMM1.

LMM2: Participants' self-reported score on partner's cooperativeness is the dependent variable; fixed effects include an intercept, the main effects of group (adolescents or adults), the order of the sessions (regarding the partner's cooperation involved fixed 78% cooperation probability, followed by shifting into 20%, 80%, and 20% for each 20, or vice versa), the interaction of group ×order. Gender (male and female) and timing (trial number from 1 to 120) were also included as the control variables. Random effects include correlated random slopes of group, gender, timing, and random intercept for participants. The group, previous trial, partner's choice, and gender are the category variables. The trial number is a continuous variable. See **Appendix 1—table 3** for the statistical results of LMM2.

## Behavioral modeling

We systematically developed models based on various assumptions regarding participants' decision-making processes in the rPDG.

### Model 1: the baseline model

We modeled each participant's choices in each trial (i.e. whether to cooperate) as outcomes from a Bernoulli distribution, where the cooperation probability is controlled by a parameter, $b \in [0, 1]$. For each participant, the probabilities of cooperation ($q(\text{cooperation})$) and defection ($q(\text{defection})$) are denoted as follows:

$$q(\text{cooperation}) = b \tag{1}$$
$$q(\text{defection}) = 1 - b \tag{2}$$

### Model 2: win-stay and loss-shift model

The model assumes that individuals adopt a tit-for-tat strategy in decision-making. Participants are likely to repeat their previous choice with a probability of $1 - \frac{\varepsilon}{2}$ if they won, and $\frac{\varepsilon}{2}$ if they lost in the last trial, where $\varepsilon$ represents the choice variability. Winning and losing are defined based on the payoff outcomes of 4 or 6 (win) and 0 or 2 (loss), respectively.

$$q_{t+1} = q_t \left(1 - \frac{\varepsilon}{2}\right) \delta + q_t \left(\frac{\varepsilon}{2}\right)(1 - \delta) \tag{3}$$

where $q_t$ denotes the probability of repeating the previous choice and $1 - q_t$ denotes the probability of shifting to another option at trial $t$.

### Model 3: reward learning model

This model assumes that participants make decisions by comparing the values of choosing cooperation and defection. The values of the two options are updated using an RL algorithm:

$$V_{c_{t+1}} = V_{c_t} + \alpha(R_t - V_{c_t}) \tag{4}$$
$$V_{d_{t+1}} = V_{d_t} + \alpha(R_t - V_{d_t}) \tag{5}$$

where $V_c$ ($V_d$) denotes the value of cooperation (defection) option. $R$ represents the reward feedback, which can be 0, 2, 4, or 6, depending on the payoff matrix. $R_t - V_{c_t}$ ($R_t - V_{d_t}$) represents the reward prediction error for the cooperation (defection) option, and $\alpha$ is the learning rate. Participants' choices are modeled by a softmax function:

$$q(\text{cooperate})_t = \frac{1}{1 + e^{\beta(V_{d_t} - V_{c_t})}} \tag{6}$$

where $q_t$ denotes the participants' probability of cooperation and $\beta$ denotes the inverse temperature. The lower the value of the inverse temperature, the greater the sensitivity to the different values between options.

## Model 4: inequality aversion model

The model assumes that participants' decisions aim to reduce both disadvantageous and advantageous inequality between themselves and their partners:

$$U_{c_t} = c_{\text{self}} - \varphi \max(c_{\text{other}} - c_{\text{self}}, 0) - \nu \max(c_{\text{self}} - c_{\text{other}}, 0) \tag{7}$$

$$U_{d_t} = d_{\text{self}} - \varphi \max(d_{\text{other}} - d_{\text{self}}, 0) - \nu \max(d_{\text{self}} - d_{\text{other}}, 0) \tag{8}$$

where $U_c$ ($U_d$) denotes the utility of cooperation (defection). $\varphi$ represents aversion to disadvantageous inequality and $\nu$ represents aversion to advantageous inequality. $c_{\text{self}}$ and $c_{\text{other}}$ denote the expected payoffs for cooperation to oneself and the partner, respectively, while $d_{\text{self}}$ and $d_{\text{other}}$ denote the expected payoffs for defection to oneself and the partner. $p$ denotes participants' partner cooperation expectation. The model assumes that participants did not update the inferred cooperation probability based on feedback; $p$ is fixed at 0.5.

Based on the payoff matrix, the payoffs for participants and their partners are calculated using the following functions:

$$c_{\text{self}} = 4p \tag{9}$$

$$c_{\text{other}} = 6 - 2p \tag{10}$$

$$d_{\text{self}} = 4p + 2 \tag{11}$$

$$d_{\text{other}} = 2 - 2p \tag{12}$$

Participants' choices are modeled by a softmax function:

$$q(\text{cooperate})_t = \frac{1}{1 + e^{\beta(U_{d_t} - U_{c_t})}} \tag{13}$$

where $q_t$ denotes the participants' probability of cooperation and $\beta$ denotes the inverse temperature. The lower the value of the inverse temperature, the greater the randomness in decisions.

## Model 5: social reward model

The model assumes that participants make decisions by comparing the expected payoff of cooperation and defection based on the payoff matrix and an additional subjective bonus from cooperation:

$$U_{c_t} = p(4 + \omega) \tag{14}$$

$$U_{c_t} = 4p + 2 \tag{15}$$

where $\omega$ represents an additional social reward associated with cooperation.

## Model 6: social reward model with RL algorithm

The model, building on Model 5, assumed that participants update their expectations of partner cooperation trial-by-trial, based on the partner's previous decisions, using a basic RL algorithm:

$$U_{c_t} = p_t(4 + \omega) \tag{16}$$

$$U_{d_t} = 4p_t + 2 \tag{17}$$

where $p_t$ denotes participants' expectation of partner cooperation probability at trial $t$ and is updated by the following function:

$$p_{t+1} = p_t + \alpha(P_t - p_t) \tag{18}$$

where $\alpha$ is the learning rate applied to the prediction error, $(P_t - p_t)$ represents the partner's decision at trial $t$, equating to 1 if the partner cooperates and 0 if the partner defects.

## Model 7: social reward model with influence model

The model is based on Model 6 and includes an additional assumption that participants update their expectation of the partner's cooperation by considering not only the partner's previous decisions but also the influence of their own previous decisions on the partner's subsequent decisions. This aspect is referred to as second-order belief and is updated by the following function:

$$p_{t+1} = p_t + \alpha(P_t - p_t) + \kappa(Q_t - q'_t) \tag{19}$$

$$q'_t = \frac{2}{\omega} \ln\left(\frac{1}{p_t} - 1\right) \frac{1}{\beta\omega} \tag{20}$$

where $Q_t$ represents the participants' decision at trial $t$, equating to 1 if the participants cooperate and 0 if participants defect. $q'_t$ represents the participants' inferred cooperation probability of themselves from the partner's perspective in trial $t$, which was inferred from function 13. Therefore, $(Q_t - q'_t)$ denotes the second-order prediction error, and $\kappa$ is the second-order learning rate that governs the updating of second-order belief.

## Model 8: social reward model with asymmetric RL rule

The model, based on Model 6, assumes that participants asymmetrically update positive expectation errors (better than expected) and negative prediction errors (worse than expected) using two distinct learning rates:

$$p_{t+1} = p_t + \alpha_+ \delta(PE) + \alpha_-(1 - \delta)PE \tag{21}$$

$$PE = P_t - p_t \tag{22}$$

$$\delta = \begin{cases} 1, & \text{if } PE > 0 \\ 0, & \text{if } PE < 0 \end{cases} \tag{23}$$

## Model fitting and model comparison

We used maximum likelihood estimation to fit models to each participant's choices across all trials. The likelihood function, based on the binomial distribution, captured the association between each participant's choices and each model's predictions. To minimize the negative log-likelihood, we employed MATLAB's (MathWorks) *fmincon* function. To enhance the likelihood of finding the global minimum, we repeated the parameter search process 500 times, using different starting points. In addition, we tested Model 9 (social reward model with Pearce–Hall learning, i.e., dynamic learning rate; see Appendix Analysis for details; and also see *Appendix 1—figure 6*).

For model evaluation, we first used the AICc, which accounts for the model's complexity and the number of observed data points (*Hurvich and Tsai, 1989*). The second metric was the protected exceedance probability from group-level Bayesian model selection (*Rigoux et al., 2014*), providing a measure of the likelihood that a specific model is superior to other models under consideration. We chose to use the AIC as the metric of goodness-of-fit for model comparison for the following statistical reasons. First, BIC is derived based on the assumption that the 'true model' must be one of the models in the limited model set one compares (*Burnham and Anderson, 2002*; *Gelman and Shalizi, 2013*), which is unrealistic in our case. In contrast, AIC does not rely on this unrealistic 'true model' assumption and instead selects out the model that has the highest predictive power in the model set (*Gelman et al., 2014*). Second, AIC is also more robust than BIC for finite sample size (*Vrieze, 2012*).

The log-likelihood is calculated as the following function:

$$L = \sum_{t=1}^{T} \log(q_t | \alpha+, \alpha-, \omega, \beta) \tag{24}$$

where $q_t$ represents the probability of participants' decision at trial $t$, equating to $q$(cooperation) if participants cooperate and $q$(defection) if participants defect.

## Model identifiability and parameter recovery analyses

We performed a model identifiability analysis to ensure that model comparisons were not compromised by model misidentification. For each model, we generated synthetic datasets using parameters estimated from the data of all participants. We then fitted each alternative model to its corresponding synthetic dataset and identified the best-fitting model through model comparison. To test robustness, we repeated this procedure 100 times, calculating the percentage of instances where each model was recognized as the best model across all synthetic datasets generated by that specific model. Consistently high percentages indicated model identifiability. Additionally, we assessed parameter recovery for the best-fitting model (model 8: social reward model with an asymmetric RL rule). This assessment involved calculating the Pearson correlation between the parameters estimated from the 100 synthetic datasets (recovered parameters) and the parameters used to generate these datasets. A higher correlation coefficient between the recovered and the estimated parameters suggested non-redundancy in the parameter space (*Appendix 1—figure 3*).

## Hidden mental variables analysis

LMM1: Participants' expectation of partner's cooperation probability that estimated from the winning model, the variable $p$, is the dependent variable. The fixed and random effects remain the same as GLMM1. See *Appendix 1—table 2* for the statistical results of LMM1.

LMM3: Participants' intrinsic reward for reciprocity that is estimated from the winning model, $p \times \omega$, are the dependent variable. The fixed and random effects remain the same as GLMM1. See *Appendix 1—table 4* for the statistical results of LMM3.

## Acknowledgements

This project has received funding from the Brain Science and Brain-like Intelligence Technology - National Science and Technology Major Project (2021ZD0200500), the National Natural Science Foundation of China (32441109, 32271092, 32130045), the National Social Science Foundation (25VRC015), the Open Research Fund of the State Key Laboratory of Cognitive Neuroscience and Learning (CNLYB2404), the Beijing Major Science and Technology Project under Contract (Z241100001324005), and the Opening Project of the State Key Laboratory of General Artificial Intelligence (SKLAGI20240P06). We thank Christian C Ruff and Xiangjuan Ren for insightful discussions.

## Additional information

### Funding

| Funder | Grant reference number | Author |
|---|---|---|
| Brain Science and Brain-like Intelligence Technology - National Science and Technology Major Project | 2021ZD0200500 | Chao Liu |
| National Natural Science Foundation of China | 32441109 | Chao Liu |
| National Social Science Foundation | 25VRC015 | Chao Liu |
| Open Research Fund of the State Key Laboratory of Cognitive Neuroscience and Learning | CNLYB2404 | Chao Liu |
| Beijing Major Science and Technology Project | Z241100001324005 | Chao Liu |

| Funder | Grant reference number | Author |
| --- | --- | --- |
| Open Research Fund of the State Key Laborary of General Artificial Intelligence | SKLAGI20240P06 | Chao Liu |
| National Natural Science Foundation of China | 32130045 | Chao Liu |
| National Natural Science Foundation of China | 32271092 | Chao Liu |

The funders had no role in study design, data collection and interpretation, or the decision to submit the work for publication.

## Author contributions

Xiaoyan Wu, Conceptualization, Resources, Data curation, Software, Formal analysis, Validation, Investigation, Visualization, Methodology, Writing – original draft, Project administration, Writing – review and editing; Hongyu Fu, Methodology, Writing – original draft, Writing – review and editing; Gökhan Aydogan, Writing – review and editing; Chunliang Feng, Validation, Visualization, Methodology, Writing – original draft, Writing – review and editing; Shaozheng Qin, Conceptualization, Resources, Writing – review and editing; Yi Zeng, Resources, Validation; Chao Liu, Resources, Supervision, Funding acquisition, Writing – review and editing

## Author ORCIDs

Xiaoyan Wu ![ORCID] https://orcid.org/0000-0002-2683-3807
Hongyu Fu ![ORCID] https://orcid.org/0009-0003-4582-4230
Shaozheng Qin ![ORCID] https://orcid.org/0000-0002-1859-2150
Chao Liu ![ORCID] https://orcid.org/0000-0003-1149-2314

## Ethics

This study was approved by the Ethics Committee of Beijing Normal University (Approval Nos. CNL_A_0001_009 and RB_A_0003_202001). Written informed consent was obtained from all adult participants and from both adolescent participants and their legal guardians prior to participation. The study was conducted in accordance with the Declaration of Helsinki. No identifiable personal information is included in this manuscript.

Reviewer #1 (Public review): https://doi.org/10.7554/eLife.106840.4.sa1
Reviewer #2 (Public review): https://doi.org/10.7554/eLife.106840.4.sa2
Author response https://doi.org/10.7554/eLife.106840.4.sa3

# Additional files

## Supplementary files

MDAR checklist

## Data availability

All data and analysis code required to reproduce the main results are publicly available at Zenodo (https://doi.org/10.5281/zenodo.15046430; *Wu, 2026*). The source code is maintained at GitHub (https://github.com/xiaoyanwu2024/Adolescents_SelfInterest_Cooperation).

The following dataset was generated:

| Author(s) | Year | Dataset title | Dataset URL | Database and Identifier |
| --- | --- | --- | --- | --- |
| Wu X | 2025 | The Self-Interest of Adolescents Overrules Cooperation in Social Dilemmas | https://doi.org/10.5281/zenodo.15046430 | Zenodo, 10.5281/zenodo.15046430 |

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

# Appendix 1

## Analysis

## Model 9: Social reward model with Pearce–Hall (PH) learning algorithm

To examine whether trial-by-trial learning rate adaptation improves model performance, we extended the social reward model by incorporating a Pearce–Hall (PH) dynamic learning rule (*Pearce and Hall, 1980*; *Li et al., 2011*). In this model, participants update their belief about their partner's cooperation probability ($\hat{r}_t$) using a learning rate $\alpha_t$ that changes adaptively as a function of recent prediction errors:

$$\hat{r}_{t+1} = \hat{r}_t + \alpha_t (r_t - \hat{r}_t), \quad \alpha_{t+1} = \alpha_t + \lambda(|PE_t| - \alpha_t),$$

where $r_t$ denotes the observed partner cooperation (or reward), $PE_t = r_t - \hat{r}_t$, and $\lambda$ controls the rate at which $\alpha_t$ adapts to the magnitude of recent prediction errors. This formulation allows the learning rate to increase following surprising outcomes and decrease when outcomes are predictable, independent of the valence of feedback. As in previous models, decision values were computed based on the expected utilities of cooperation ($U^{coop} = \hat{r}_t(4 + \omega)$) and defection ($U^{def} = \hat{r}_t \times 4 + 2$), and choice probabilities were obtained via a softmax function governed by the inverse temperature parameter $\beta$. Model comparison indicated that the PH dynamic learning model did not outperform the best-fitting asymmetric RL model (Model 8) in either age group (see *Appendix 1—figure 6*).

## Hierarchical Bayesian estimation

To complement the maximum likelihood estimation analyses, we additionally implemented a hierarchical Bayesian estimation for the best-fitting model. The hierarchical model incorporated both group-level (adolescent and adult) and individual-level structures to improve the stability and identifiability of parameter estimation.

At the individual level, each participant $s$ was characterized by four parameters: the positive learning rate ($\alpha+_s$), negative learning rate ($\alpha-_s$), social reward weight ($\omega_s$), and inverse temperature ($\beta_s$). Each parameter was assumed to follow a normal distribution centered on the corresponding group-level mean with a group-specific standard deviation:

$$\begin{aligned}
\alpha_s^+ &\sim \mathcal{N}(\mu_{\alpha_{g[s]}^+}, \sigma_{\alpha_{g[s]}^+}), \\
\alpha_s^- &\sim \mathcal{N}(\mu_{\alpha_{g[s]}^-}, \sigma_{\alpha_{g[s]}^-}), \\
\omega_s &\sim \mathcal{N}(\mu_{\omega_{g[s]}}, \sigma_{\omega_{g[s]}}), \\
\beta_s &\sim \mathcal{N}(\mu_{\beta_{g[s]}}, \sigma_{\beta_{g[s]}}),
\end{aligned}$$

where $g[s] \in \{\text{adolescent}, \text{adult}\}$ denotes the group of participant $s$.

At the group level, the hyperparameters μ and σ were assigned weakly informative priors to regularize estimation while allowing flexibility across age groups. The likelihood for each trial $t$ followed a Bernoulli distribution based on the model-predicted cooperation probability $p_{s,t}$, computed via the softmax (logistic) transformation of the utility difference:

$$y_{s,t} \sim \text{Bernoulli}(p_{s,t}), \quad p_{s,t} = \frac{1}{1 + \exp[-\beta_s(U_{s,t}^{coop} - U_{s,t}^{def})]},$$

where $U_{s,t}^{coop}$ and $U_{s,t}^{def}$ denote the expected utilities of cooperation and defection, respectively, updated trial-by-trial using asymmetric learning rates:

$$PE_{s,t} = r_{s,t} - \hat{r}_{s,t-1}, \quad \hat{r}_{s,t} = \begin{cases} \hat{r}_{s,t-1} + \alpha_s^+ \cdot PE_{s,t}, & \text{if } PE_{s,t} > 0, \\ \hat{r}_{s,t-1} + \alpha_s^- \cdot PE_{s,t}, & \text{if } PE_{s,t} < 0. \end{cases}$$

The hierarchical Bayesian model was implemented in Stan (*Stan Development Team, 2023*) and fitted using the `sampling()` function in `RStan`. Four independent Markov chains were run with 4000 iterations each, including 1000 warm-up samples for adaptation. Convergence was evaluated using the potential scale reduction statistic $\hat{R}$, with all parameters showing $\hat{R} < 1.01$, indicating good

convergence and stable sampling (see *Appendix 1—figure 7a*). Trace plots for the group-level parameters ($\alpha+$, $\alpha-$, $\omega$, and $\beta$) further confirmed well-mixed chains (see *Appendix 1—figure 7b*).

Posterior group-level parameter estimates ($\mu$ parameters) were then directly compared between adolescents and adults. The hierarchical Bayesian results closely replicated those obtained from the MLE approach, demonstrating consistent age-group differences in parameters (see *Appendix 1—figure 5*).

## Robustness analyses and extensions of main findings

To further examine the robustness of the findings, we first reconduct GLMM1 and LMM1–3 by replacing the categorical Group factor (adolescents vs. adults) with a continuous age predictor, yielding $GLMM_{sup}1$ and $LMM_{sup}1$-3. Overall, the patterns replicated those observed for the Group effect: the Previous trial ×Partner's choice ×Age interaction was significant in $GLMM_{sup}1$ ($b$=0.02, 95% CI = [0.001, 0.036], $p$=0.038, see *Appendix 1—table 5*) and $LMM_{sup}3$ ($b$=0.04, 95% CI = [0.031, 0.046], p<0.001, see *Appendix 1—table 8*), and marginally significant in $LMM_{sup}1$ ($b$=0.0012, 95% CI = $[-1.63 \times 10^{-7}$, 0.002], p=0.050, see *Appendix 1—table 6*). Consistent with the nonsignificant effect of the group in LMM2 (see *Appendix 1—table 3*), the corresponding age effect in $LMM_{sup}2$ (see *Appendix 1—table 7*) was also not significant ($b$=0.03, 95% CI = [–0.168, 0.222], $p$=0.784). Furthermore, based on GLMM1, LMM1, and LMM3, we further included the phase factor (stable versus changing phase) as a fixed effect with random slopes, yielding $GLMM_{sup}2$ and $LMM_{sup}4$–5. This specification accounts for potential effects of the experimental manipulation of the partner's cooperation, which differed between the first and second halves of the trials. We also replicated the main results: the Previous trial ×Partner's choice ×Age interaction was significant in $GLMM_{sup}2$ ($b$=0.25, 95% CI = [0.128, 0.366], p<0.001, see *Appendix 1—table 9*), $LMM_{sup}4$ ($b$=0.03, 95% CI = [0.022, 0.037], p<0.001, see *Appendix 1—table 10*), and $LMM_{sup}5$ ($b$=0.37, 95% CI = [0.314, 0.417], p<0.001, see *Appendix 1—table 11*). Besides, to account for the potential influence of individual differences in prosociality on participants' cooperative behavior and the reward for reciprocity, we also extended GLMM1 and LMM3 by adding the measured SVO as a fixed effect with random slopes, yielding $GLMM_{sup}3$ and $LMM_{sup}6$. The results showed that higher SVO positively predicted greater cooperation ($b$=0.02, 95% CI = [0.007, 0.027], p<0.001, see *Appendix 1—table 12*), whereas its effect on the reward for reciprocity was not significant ($b$=0.006, 95% CI = [–0.002, 0.014], p=0.137, see *Appendix 1—table 13*). Importantly, the primary findings remained unchanged after controlling for SVO.

Finally, we conducted an exploratory analysis to examine the relationship between participants' expectations of partner's cooperation probability and their own cooperative behavior. Specifically, we estimated $GLMM_{sup}4$, in which participants' choices served as the dependent variable. The fixed effects model included trial number, gender, group, cooperation expectation, and the interaction between group and cooperation expectation. Random effects included correlated random slopes of trial number, gender, group, cooperation expectation, and a random intercept for participants. We showed that participants' cooperation expectations positively predicted cooperative behavior ($b$=7.90, 95% CI = [7.258, 8.459], p<0.001, see *Appendix 1—table 14*), indicating that their cooperation depends on their estimates of partner' belief. In addition, we found the interaction between group and cooperation expectation was not significant ($b$=0.015, 95% CI = [–0.358, 0.387], $p$=0.938), suggesting that this social learning process likely operates similarly in adolescents and adults and is not driven by age-group differences.

**Appendix 1—table 1.** Statistical results for cooperation decision (GLMM1).
Source data available at https://raw.githubusercontent.com/xiaoyanwu2024/Adolescents_SelfInterest_Cooperation/main/data/dataset.csv.

| Fixed effects | Estimated beta value | SE | t value | p value |
|---|---|---|---|---|
| (Intercept) | 0.47 | 0.22 | 2.10 | p=0.036 |
| Timing | –0.001 | 0.001 | –1.01 | p=0.311 |
| Gender | 0.14 | 0.15 | 0.93 | p=0.355 |
| Group | 0.79 | 0.24 | 3.23 | p=0.001 |

*Appendix 1—table 1 Continued on next page*

*Appendix 1—table 1 Continued*

| Fixed effects | Estimated beta value | SE | t value | p value |
|---|---|---|---|---|
| Previous trial | −1.81 | 0.08 | −23.43 | p<0.001 |
| Partner's choice | −0.73 | 0.10 | −7.21 | p<0.001 |
| Group × Previous trial | −0.41 | 0.10 | −3.89 | p<0.001 |
| Group × Partner's choice | −0.19 | 0.11 | −1.77 | p=0.076 |
| Previous trial × Partner's choice | 1.12 | 0.04 | 25.43 | p<0.001 |
| Group × Previous trial × Partner's choice | 0.24 | 0.06 | 4.05 | p<0.001 |

Coding of variables. Trial number: integer sequence from 2 to 120; gender: female = 0, male = 1; group: adolescents = 0, adults = 1; previous trials: last one trial = 1, last two trials = 2, last three trials = 3; partner's choice: cooperation = 1, defection = 0.

**Appendix 1—table 2.** Statistical results for partner cooperation expectation (LMM1).
Source data available at https://raw.githubusercontent.com/xiaoyanwu2024/Adolescents_SelfInterest_Cooperation/main/data/dataset.csv.

| Fixed effects | Estimated beta value | SE | t value | p value |
|---|---|---|---|---|
| (Intercept) | 0.59 | 0.03 | 17.52 | p<0.001 |
| Trial number | <0.001 | <0.001 | 0.02 | p=0.982 |
| Gender | −0.01 | 0.03 | −0.36 | p=0.718 |
| Group | −0.04 | 0.03 | −1.29 | p=0.198 |
| Previous trial | −0.25 | 0.01 | −48.97 | p<0.001 |
| Partner's choice | −0.06 | 0.01 | −4.90 | p<0.001 |
| Group × Previous trial | −0.04 | 0.01 | −6.10 | p<0.001 |
| Group × Partner's choice | −0.02 | 0.01 | −2.73 | p=0.006 |
| Previous trial × Partner's choice | 0.16 | 0.002 | 53.89 | p<0.001 |
| Group × Previous trial × Partner's choice | 0.03 | 0.004 | 7.31 | p<0.001 |

Coding of variables is consistent with *Appendix 1—table 1*.

**Appendix 1—table 3.** Statistical results for self-reported perceived partner cooperativeness (LMM2).
Source data available at https://raw.githubusercontent.com/xiaoyanwu2024/Adolescents_SelfInterest_Cooperation/main/data/SubjectiveRatingScore.csv.

| Fixed effects | Estimated beta value | SE | t value | p value |
|---|---|---|---|---|
| (Intercept) | 5.82 | 0.27 | 21.93 | p<0.001 |
| Group | 0.17 | 0.35 | 0.50 | p=0.616 |
| Gender | −0.44 | 0.11 | −3.84 | p<0.001 |
| Order of sessions | 0.09 | 0.15 | 0.60 | p=0.549 |
| Rating number | −0.13 | 0.02 | −5.55 | p<0.001 |
| Group × Order | 0.38 | 0.22 | 1.72 | p=0.085 |

Coding of variables. Group: adolescents = 0, adults = 1; gender: female = 0, male = 1; order of sessions: stable to volatile = 0, volatile to stable = 1; rating number: integer sequence from 1 to 8.

**Appendix 1—table 4.** Statistical results for intrinsic reward for reciprocity (LMM3).
Source data available at https://raw.githubusercontent.com/xiaoyanwu2024/Adolescents_SelfInterest_Cooperation/main/data/dataset.csv.

| Fixed effects | Estimated beta value | SE | t value | p value |
|---|---|---|---|---|
| (Intercept) | 2.43 | 0.14 | 17.04 | p<0.001 |
| Trial number | −0.001 | 0.001 | −0.60 | p=0.551 |
| Gender | 0.08 | 0.13 | 0.59 | p=0.554 |
| Group | 0.52 | 0.15 | 3.44 | p<0.001 |
| Previous trial | −1.23 | 0.03 | −36.18 | p<0.001 |
| Partner's choice | −0.27 | 0.08 | −3.24 | p=0.001 |
| Group × Previous trial | −0.57 | 0.05 | −11.95 | p<0.001 |
| Group × Partner's choice | −0.24 | 0.05 | −4.43 | p=0.006 |
| Previous trial × Partner's choice | 0.79 | 0.02 | 40.64 | p<0.001 |
| Group × Previous trial × Partner's choice | 0.37 | 0.02 | 13.70 | p<0.001 |

Coding of variables is consistent with **Appendix 1—table 1**.

**Appendix 1—table 5.** Statistical results for cooperation decision with age ($GLMM_{sup}1$).
Source data available at https://raw.githubusercontent.com/xiaoyanwu2024/Adolescents_SelfInterest_Cooperation/main/data/dataset.csv.

| Fixed effects | Estimated beta value | SE | t value | p value |
|---|---|---|---|---|
| (Intercept) | −1.55 | 0.73 | −2.12 | p=0.034 |
| Trial number | −0.001 | 0.001 | −0.94 | p=0.346 |
| Gender | −0.06 | 0.18 | −0.34 | p=0.732 |
| Previous trial | −0.49 | 0.15 | −3.30 | p=0.001 |
| Partner's choice | −0.60 | 0.33 | −1.83 | p=0.068 |
| Age | 0.10 | 0.04 | 2.49 | p=0.013 |
| Previous trial × Partner's choice | 0.89 | 0.17 | 5.10 | p=<0.001 |
| Previous trial × Age | −0.02 | 0.01 | −2.00 | p=0.045 |
| Partner's choice × Age | −0.01 | 0.02 | −0.68 | p=0.499 |
| Previous trial × Partner's choice × Age | 0.02 | 0.01 | 2.07 | p=0.038 |

Age was treated as a continuous variable, and all other variables were coded as in **Appendix 1—table 1**.

**Appendix 1—table 6.** Statistical results for partner cooperation expectation with age ($LMM_{sup}1$).
Source data available at https://raw.githubusercontent.com/xiaoyanwu2024/Adolescents_SelfInterest_Cooperation/main/data/dataset.csv.

| Fixed effects | Estimated beta value | SE | t value | p value |
|---|---|---|---|---|
| (Intercept) | 0.76 | 0.12 | 6.52 | p<0.001 |
| Trial number | $-2.53 \times 10^{-7}$ | 0.0002 | −0.002 | p=0.999 |
| Gender | −0.01 | 0.03 | −0.32 | p=0.746 |
| Previous trial | −0.10 | 0.01 | −10.51 | p<0.001 |
| Partner's choice | −0.03 | 0.03 | −1.05 | p=0.294 |
| Age | −0.014 | 0.006 | −2.27 | p=0.023 |
| Previous trial × Partner's choice | 0.15 | 0.01 | 13.24 | p<0.001 |
| Previous trial × Age | $-4.76 \times 10^{-7}$ | 0.0004 | 0.001 | p=0.999 |
| Partner's choice × Age | −0.002 | 0.001 | −1.74 | p=0.082 |

*Appendix 1—table 6 Continued on next page*

*Appendix 1—table 6 Continued*

| Fixed effects | Estimated beta value | SE | t value | p value |
|---|---|---|---|---|
| Previous trial × Partner's choice × Age | 0.0012 | 0.0006 | 1.96 | p=0.050 |

Coding of variables is consistent with *Appendix 1—table 5*.

**Appendix 1—table 7.** Statistical results for self-reported perceived partner cooperativeness with age (LMM$_{sup}$2).
Source data available at https://raw.githubusercontent.com/xiaoyanwu2024/Adolescents_SelfInterest_Cooperation/main/data/SubjectiveRatingScore.csv.

| Fixed effects | Estimated beta value | SE | t value | p value |
|---|---|---|---|---|
| (Intercept) | 5.18 | 1.89 | 2.74 | p=0.006 |
| Gender | –0.37 | 0.19 | –1.96 | p=0.050 |
| Order of sessions | 2.31 | 1.21 | 1.90 | p=0.057 |
| Age | 0.03 | 0.10 | 0.27 | p=0.784 |
| Rating number | –0.13 | 0.02 | –6.65 | p<0.001 |
| Order × Age | –0.10 | 0.07 | –1.57 | p=0.116 |

Age was treated as a continuous variable, and all other variables were coded as in *Appendix 1—table 3*.

**Appendix 1—table 8.** Statistical results for partner cooperation expectation with age (LMM$_{sup}$3).
Source data available at https://raw.githubusercontent.com/xiaoyanwu2024/Adolescents_SelfInterest_Cooperation/main/data/dataset.csv.

| Fixed effects | Estimated beta value | SE | t value | p value |
|---|---|---|---|---|
| (Intercept) | 2.02 | 0.58 | 3.49 | p<0.001 |
| Trial number | –0.001 | 0.001 | –0.60 | p=0.546 |
| Gender | –0.07 | 0.13 | –0.50 | p=0.619 |
| Previous trial | –0.22 | 0.06 | –3.59 | p<0.001 |
| Partner's choice | –0.11 | 0.18 | –0.62 | p=0.533 |
| Age | 0.02 | 0.03 | 0.61 | p=0.540 |
| Previous trial × Partner's choice | 0.25 | 0.08 | 3.28 | p=0.001 |
| Previous trial × Age | –0.02 | 0.003 | –5.24 | p<0.001 |
| Partner's choice × Age | –0.02 | 0.01 | –1.85 | p=0.065 |
| Previous trial × Partner's choice × Age | 0.04 | 0.004 | 9.82 | p<0.001 |

Coding of variables is consistent with *Appendix 1—table 5*.

**Appendix 1—table 9.** Statistical results for cooperation decision with phase (GLMM$_{sup}$2).
Source data available at https://raw.githubusercontent.com/xiaoyanwu2024/Adolescents_SelfInterest_Cooperation/main/data/dataset.csv.

| Fixed effects | Estimated beta value | SE | t value | p value |
|---|---|---|---|---|
| (Intercept) | 1.06 | 0.24 | 4.34 | p<0.001 |
| Trial number | –0.01 | 0.001 | –4.46 | p<0.001 |
| Gender | 0.17 | 0.15 | 1.10 | p=0.271 |
| Group | 0.60 | 0.19 | 3.15 | p=0.002 |
| Previous trial | –0.64 | 0.04 | –16.49 | p<0.001 |
| Partner's choice | –0.75 | 0.10 | –7.73 | p<0.001 |

*Appendix 1—table 9 Continued on next page*

*Appendix 1—table 9 Continued*

| Fixed effects | Estimated beta value | SE | t value | p value |
|---|---|---|---|---|
| Phase | −0.66 | 0.10 | −6.45 | p<0.001 |
| Group × Previous trial | −0.16 | 0.05 | −3.15 | p=0.002 |
| Group × Partner's choice | −0.19 | 0.11 | −1.77 | p=0.077 |
| Previous trial × Partner's choice | 1.04 | 0.04 | 23.44 | p<0.001 |
| Group × Previous trial × Partner's choice | 0.25 | 0.06 | 4.07 | p<0.001 |

Phase was dummy-coded (0=stable, 1=changing), and all other variables were coded as in *Appendix 1—table 1*.

**Appendix 1—table 10.** Statistical results for partner cooperation expectation with phase (LMM$_{sup}$4). Source data available at https://raw.githubusercontent.com/xiaoyanwu2024/Adolescents_SelfInterest_Cooperation/main/data/dataset.csv.

| Fixed effects | Estimated beta value | SE | t value | p value |
|---|---|---|---|---|
| (Intercept) | 0.77 | 0.03 | 28.09 | p<0.001 |
| Trial number | $-5.86 \times 10^{-6}$ | $9.25 \times 10^{-5}$ | −0.06 | p=0.949 |
| Gender | −0.02 | 0.03 | −0.96 | p=0.337 |
| Group | −0.06 | 0.03 | −2.36 | p=0.018 |
| Previous trial | −0.08 | 0.002 | −33.62 | p<0.001 |
| Partner's choice | −0.07 | 0.01 | −6.65 | p<0.001 |
| Phase | −0.14 | 0.01 | −19.16 | p<0.001 |
| Group × Previous trial | −0.013 | 0.003 | −4.02 | p<0.001 |
| Group × Partner's choice | −0.025 | 0.007 | −3.11 | p=0.002 |
| Previous trial × Partner's choice | 0.15 | 0.003 | 50.67 | p<0.001 |
| Group × Previous trial × Partner's choice | 0.03 | 0.004 | 7.40 | p<0.001 |

Coding of variables is consistent with *Appendix 1—table 9*.

**Appendix 1—table 11.** Statistical results for intrinsic reward for reciprocity with phase (LMM$_{sup}$5). Source data available at https://raw.githubusercontent.com/xiaoyanwu2024/Adolescents_SelfInterest_Cooperation/main/data/dataset.csv.

| Fixed effects | Estimated beta value | SE | t value | p value |
|---|---|---|---|---|
| (Intercept) | 3.57 | 0.14 | 25.57 | p<0.001 |
| Trial number | −0.0011 | 0.0006 | −2.01 | p=0.045 |
| Gender | 0.02 | 0.13 | 0.18 | p=0.860 |
| Group | 0.25 | 0.13 | 1.96 | p=0.050 |
| Previous trial | −0.38 | 0.02 | −22.73 | p<0.001 |
| Partner's choice | −0.33 | 0.07 | −4.59 | p<0.001 |
| Phase | −0.83 | 0.06 | −14.76 | p<0.001 |
| Group × Previous trial | −0.19 | 0.02 | −8.79 | p<0.001 |
| Group × Partner's choice | −0.27 | 0.05 | −5.18 | p<0.001 |
| Previous trial × Partner's choice | 0.70 | 0.02 | 37.41 | p<0.001 |
| Group × Previous trial × Partner's choice | 0.37 | 0.03 | 13.99 | p<0.001 |

Coding of variables is consistent with *Appendix 1—table 9*.

**Appendix 1—table 12.** Statistical results for cooperation decision with SVO (GLMM$_{sup}$3). Source data available at https://raw.githubusercontent.com/xiaoyanwu2024/Adolescents_SelfInterest_Cooperation/main/data/dataset.csv.

| Fixed effects | Estimated beta value | SE | t value | p value |
|---|---|---|---|---|
| (Intercept) | –0.63 | 0.22 | –2.88 | p=0.004 |
| Trial number | –0.002 | 0.001 | –1.16 | p=0.247 |
| Gender | 0.18 | 0.14 | 1.27 | p=0.202 |
| Group | 0.56 | 0.18 | 3.11 | p=0.002 |
| Previous trial | –0.71 | 0.04 | –18.17 | p<0.001 |
| Partner's choice | –0.77 | 0.10 | –7.41 | p<0.001 |
| SVO | 0.02 | 0.005 | 3.35 | p<0.001 |
| Group ×Previous trial | –0.14 | 0.05 | –2.84 | p=0.005 |
| Group ×Partner's choice ×Age | –0.16 | 0.11 | –1.44 | p=0.150 |
| Previous trial ×Partner's choice | 1.13 | 0.04 | 25.18 | p<0.001 |
| Group ×Previous trial ×Partner's choice | 0.25 | 0.06 | 4.06 | p<0.001 |

SVO was treated as a continuous variable, and all other variables were coded as in **Appendix 1—table 1**.

**Appendix 1—table 13.** Statistical results for intrinsic reward for reciprocity with SVO (LMM$_{sup}$6). Source data available at https://raw.githubusercontent.com/xiaoyanwu2024/Adolescents_SelfInterest_Cooperation/main/data/dataset.csv.

| Fixed effects | Estimated beta value | SE | t value | p value |
|---|---|---|---|---|
| (Intercept) | 2.03 | 0.15 | 13.73 | p<0.001 |
| Trial number | –0.001 | 0.00 1 | –0.71 | p=0.479 |
| Gender | 0.02 | 0.12 | 0.18 | p=0.855 |
| Group | 0.26 | 0.13 | 2.07 | p=0.039 |
| Previous trial | –0.45 | 0.02 | –25.75 | p<0.001 |
| Partner's choice | –0.31 | 0.08 | –3.65 | p<0.001 |
| Phase | 0.006 | 0.004 | 1.49 | p=0.137 |
| Group × Previous trial | –0.18 | 0.02 | –8.02 | p<0.001 |
| Group × Partner's choice | –0.17 | 0.06 | –3.03 | p=0.002 |
| Previous trial × Partner's choice | 0.80 | 0.02 | 40.26 | p<0.001 |
| Group × Previous trial × Partner's choice | 0.37 | 0.03 | 13.38 | p<0.001 |

Coding of variables is consistent with **Appendix 1—table 12**.

**Appendix 1—table 14.** Statistical results for cooperation decision predicted by cooperation expectation (GLMM$_{sup}$4). Source data available at https://raw.githubusercontent.com/xiaoyanwu2024/Adolescents_SelfInterest_Cooperation/main/data/dataset.csv.

| Fixed effects | Estimated beta value | SE | t value | p value |
|---|---|---|---|---|
| (Intercept) | –4.57 | 0.35 | –12.92 | p<0.001 |
| Trial number | –0.002 | 0.001 | –2.25 | p=0.024 |
| Gender | 0.20 | 0.28 | 0.71 | p=0.475 |
| Group | 1.37 | 0.34 | 4.06 | p<0.001 |

*Appendix 1—table 14 Continued on next page*

*Appendix 1—table 14 Continued*

| Fixed effects | Estimated beta value | SE | t value | p value |
|---|---|---|---|---|
| Cooperation expectation | 7.90 | 0.33 | 24.05 | p<0.001 |
| Group × Cooperation expectation | 0.01 | 0.19 | 0.08 | p=0.938 |

Coding of variables. Trial number: integer sequence from 2 to 120; gender: female = 0, male = 1; group: adolescents = 0, adults = 1. Cooperation expectation was treated as a continuous variable.

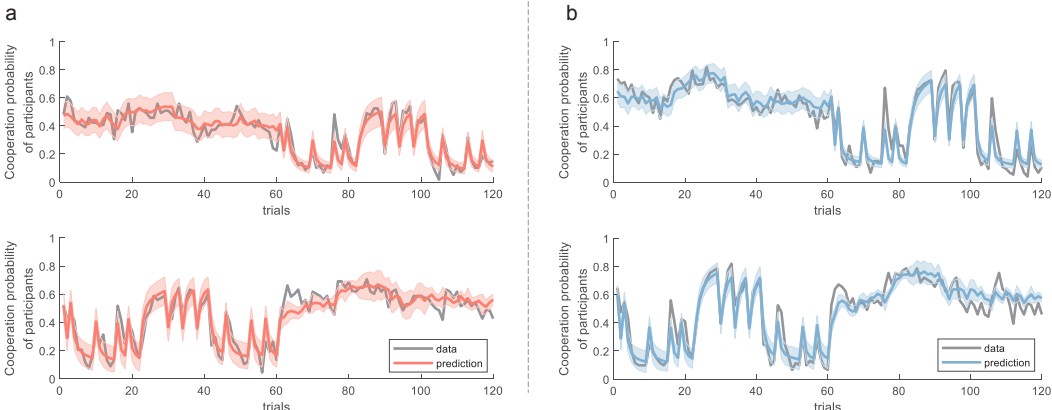

**Appendix 1—figure 1.** Model prediction. This figure compares the empirical cooperation probabilities and the model-predicted values for adults (**a**) and adolescents (**b**). The x-axis represents the trial number, and the y-axis represents the mean cooperation probability across participants. The shaded areas indicate the 95% confidence intervals.

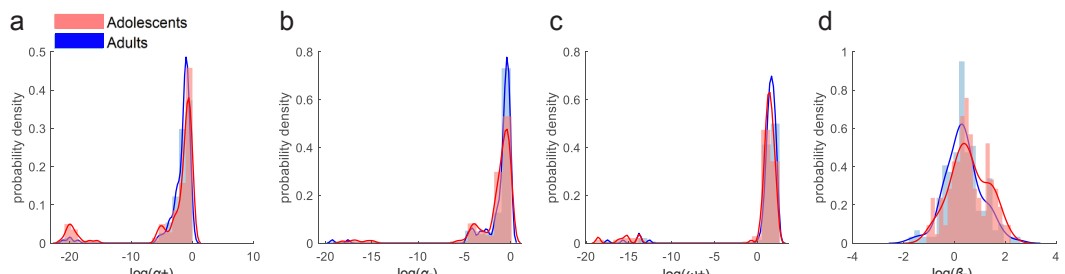

**Appendix 1—figure 2.** Distributions of estimated parameters from the best-fitting model. Each panel displays one parameter. The histograms and their kernel fits are represented by color bars and curves, respectively. Red indicates participants in the adolescent sample, and blue denotes those in the adult sample. Parameters have been transformed into a log scale for enhanced visualization.

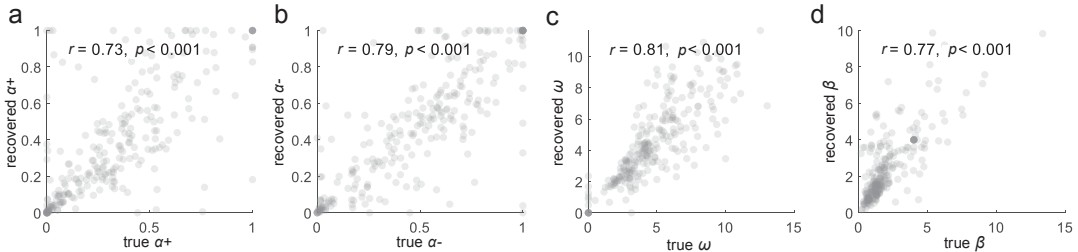

**Appendix 1—figure 3.** Parameter recovery for the best-fitting model. Each panel represents one parameter. Each dot corresponds to one virtual participant. The value of *r* indicates Pearson's correlation coefficient between the true values (estimated from the participants) and the recovered parameters.

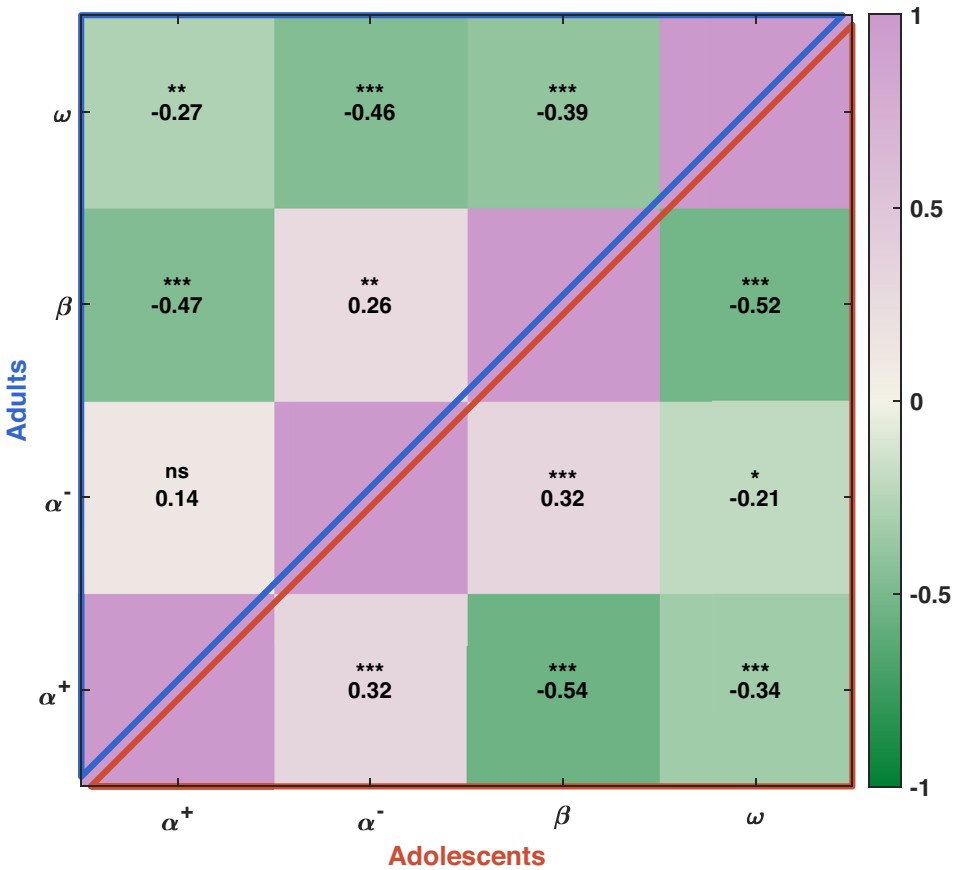

**Appendix 1—figure 4.** Partial correlation matrices among parameters for the best-fitting model. The upper-triangular cells show partial correlations for adults, and the lower-triangular cells show partial correlations for adolescents. Each cell shows the partial Pearson correlation coefficient (controlling for the other parameters). Colors range from green (negative) to violet (positive), with the color bar spanning [–1,1]. Notes: *n.s.* p > 0.05; *p < 0.05; **p < 0.01; ***p < 0.001.

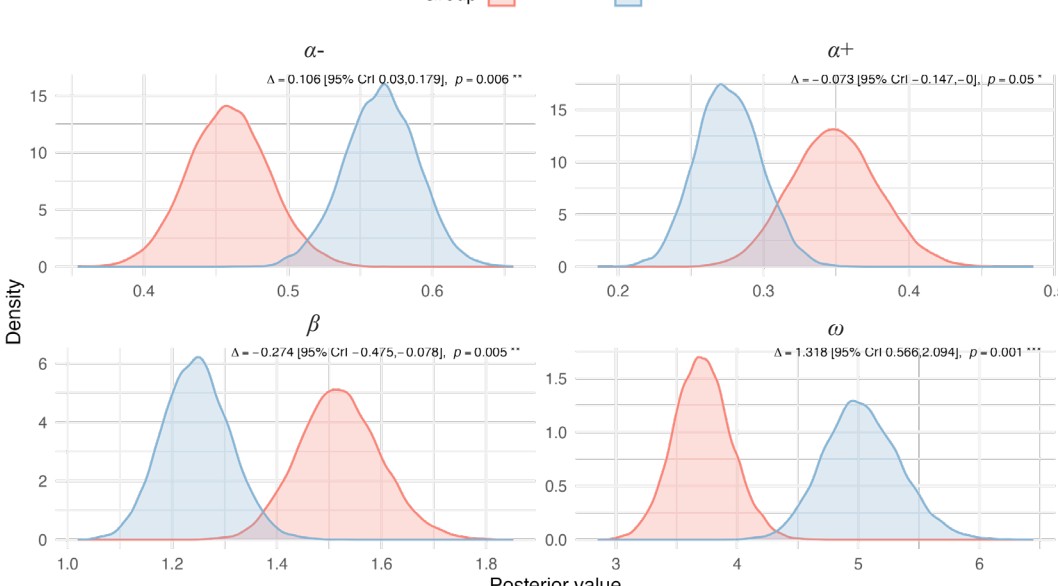

**Appendix 1—figure 5.** Group-level posterior distributions from the hierarchical Bayesian estimation for adolescents and adults. Posterior densities are shown separately for adolescents (red) and adults (blue). $\Delta$ values indicate the posterior mean difference (Adult – Adolescent) with 95% credible intervals (CrI) and Bayesian $p$ values. Compared with adolescents, adults exhibited higher positive learning rates ($\alpha+$) and lower negative learning rates ($\alpha-$), suggesting greater differentiation between learning from positive and negative feedback. Adults also showed lower inverse temperature ($\beta$), indicating more exploratory decision behavior, and higher social reward weight ($\omega$), reflecting greater valuation of reciprocity or social outcomes. Notes: *n.s.* p > 0.05; *p < 0.05; **p < 0.01; ***p < 0.001.

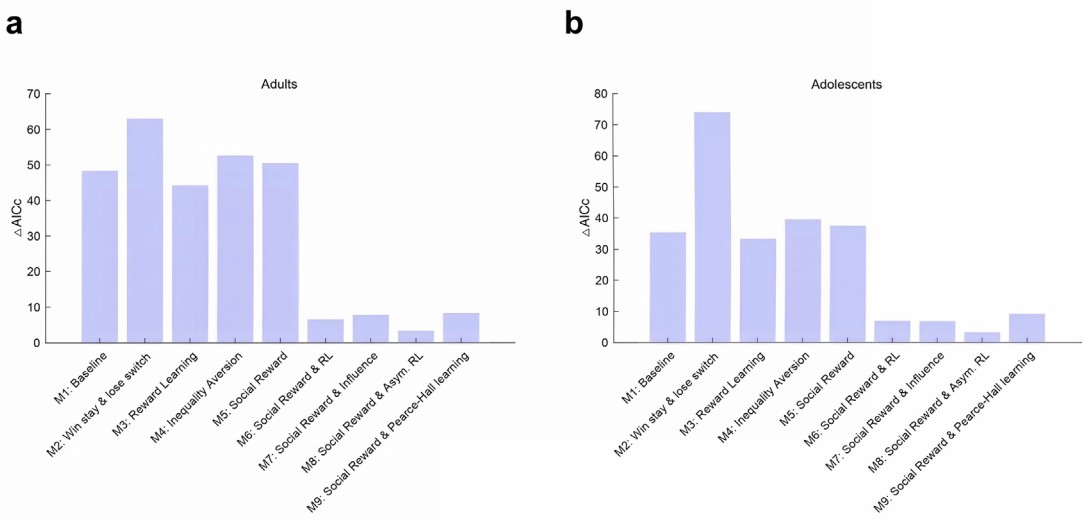

**Appendix 1—figure 6.** Model comparison results for (**a**) adults and (**b**) adolescents, including the newly added M9 (Social Reward and Pearce–Hall learning). Lower ΔAICc values indicate better model fits. The dynamic learning rate model (Model 9: Social Reward model with dynamic RL algorithm) did not outperform the best-fitting model (Model 8) in either group.

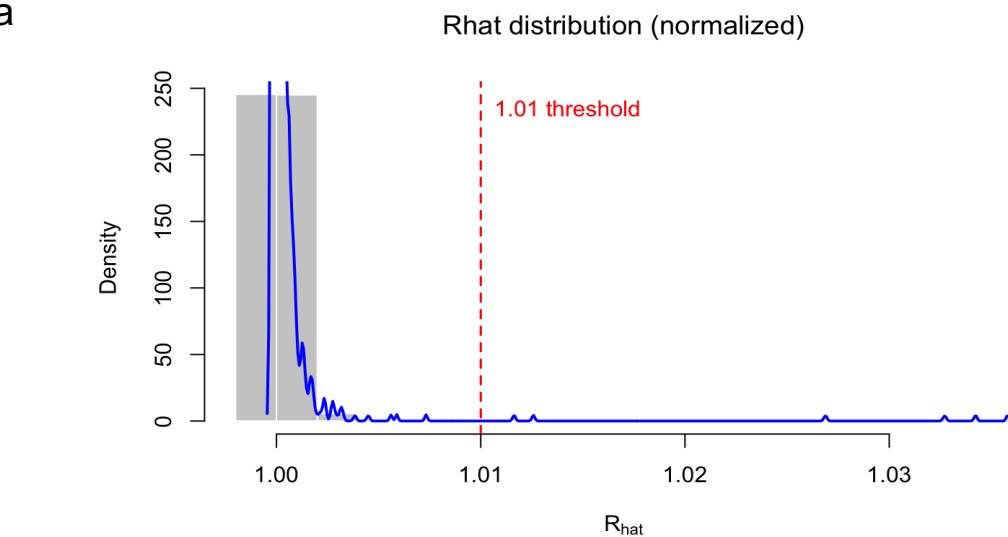

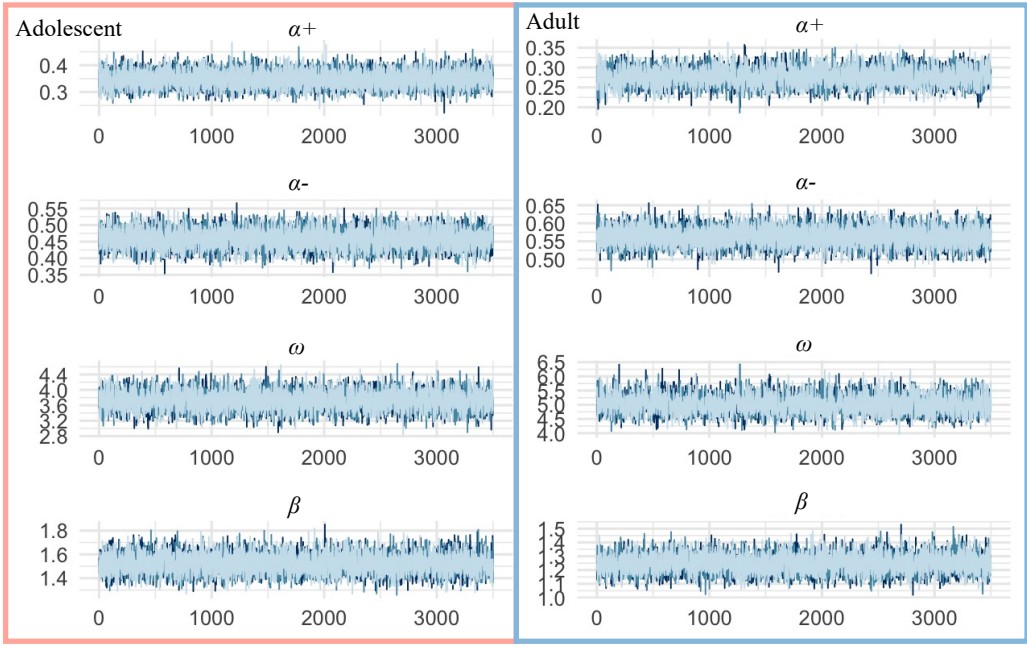

**Appendix 1—figure 7.** Convergence diagnostics for the hierarchical Bayesian model. (**a**) Distribution of $\hat{R}$ (Rhat) values across all model parameters. The majority of $\hat{R}$ values are below the conservative convergence threshold of 1.01 (red dashed line), indicating stable and well-mixed MCMC chains. The gray shaded area highlights the region where $\hat{R} \leq 1.01$. (**b**) Trace plots for the group-level parameters (four chains) in adolescents (left, red box) and adults (right, blue box). Each line represents the sampled posterior values of one chain across iterations, with overlapping traces and stable fluctuations confirming adequate convergence and mixing for all key parameters ($\alpha+$, $\alpha-$, $\omega$, $\beta$).

