## [Editor Report · eLife Assessment]

This **important** work investigates cooperative behaviors in adolescents using a repeated Prisoner's Dilemma game. The approach used in the study is **solid**. The impact of this work could be further enhanced with more rigorous modelling procedures and more modeling selection/comparison details, as well as by framing the findings in terms of the specific game-theoretic context, rather than general cooperation. Findings from this study will be of interest to developmental psychologists, economists, and social psychologists.

---

## [Referee Report · Reviewer #1 (Public review)]

Summary:

Wu and colleagues aimed to explain previous findings that adolescents, compared to adults, show reduced cooperation following cooperative behaviour from a partner in several social scenarios. The authors analysed behavioural data from adolescents and adults performing a zero-sum Prisoner's Dilemma task and compared a range of social and non-social reinforcement learning models to identify potential algorithmic differences. Their findings suggest that adolescents' lower cooperation is best explained by a reduced learning rate for cooperative outcomes, rather than differences in prior expectations about the cooperativeness of a partner. The authors situate their results within the broader literature, proposing that adolescents' behaviour reflects a stronger preference for self-interest rather than a deficit in mentalising.

Strengths:

The work as a whole suggests that, in line with past work, adolescents prioritise value accumulation, and this can be, in part, explained by algorithmic differences in weighted value learning. The authors situate their work very clearly in past literature, and make it obvious the gap they are testing and trying to explain. The work also includes social contexts which move the field beyond non-social value accumulation in adolescents. The authors compare a series of formal approaches that might explain the results and establish generative and model-comparison procedures to demonstrate the validity of their winning model and individual parameters. The writing was clear, and the presentation of the results was logical and well-structured.

Weaknesses:

I had some concerns about the methods used to fit and approximate parameters of interest. Namely, the use of maximum likelihood versus hierarchical methods to fit models on an individual level, which may reduce some of the outliers noted in the supplement, and also may improve model identifiability.

There was also little discussion given the structure of the Prisoner's Dilemma, and the strategy of the game (that defection is always dominant), meaning that the preferences of the adolescents cannot necessarily be distinguished from the incentives of the game, i.e. they may seem less cooperative simply because they want to play the dominant strategy, rather than a lower preferences for cooperation if all else was the same.

The authors have now addressed my comments and concerns in their revised version.

Appraisal & Discussion:

Overall, I believe this work has the potential to make a meaningful contribution to the field. Its impact would be strengthened by more rigorous modelling checks and fitting procedures, as well as by framing the findings in terms of the specific game-theoretic context, rather than general cooperation.

Comments on revisions:

Thank you to the authors for addressing my comments and concerns.

---

## [Referee Report · Reviewer #2 (Public review)]

Summary:

This manuscript investigates age-related differences in cooperative behavior by comparing adolescents and adults in a repeated Prisoner's Dilemma Game (rPDG). The authors find that adolescents exhibit lower levels of cooperation than adults. Specifically, adolescents reciprocate partners' cooperation to a lesser degree than adults do. Through computational modeling, they show that this relatively low cooperation rate is not due to impaired expectations or mentalizing deficits, but rather a diminished intrinsic reward for reciprocity. A social reinforcement learning model with asymmetric learning rate best captured these dynamics, revealing age-related differences in how positive and negative outcomes drive behavioral updates. These findings contribute to understanding the developmental trajectory of cooperation and highlight adolescence as a period marked by heightened sensitivity to immediate rewards at the expense of long-term prosocial gains.

Strengths:

Rigid model comparison and parameter recovery procedure. Conceptually comprehensive model space. Well-powered samples.

Weaknesses:

A key conceptual distinction between learning from non-human agents (e.g., bandit machines) and human partners is that the latter are typically assumed to possess stable behavioral dispositions or moral traits. When a non-human source abruptly shifts behavior (e.g., from 80% to 20% reward), learners may simply update their expectations. In contrast, a sudden behavioral shift by a previously cooperative human partner can prompt higher-order inferences about the partner's trustworthiness or the integrity of the experimental setup (e.g., whether the partner is truly interactive or human). The authors may consider whether their modeling framework captures such higher-order social inferences. Specifically, trait-based models-such as those explored in Hackel et al. (2015, Nature Neuroscience)-suggest that learners form enduring beliefs about others' moral dispositions, which then modulate trial-by-trial learning. A learner who believes their partner is inherently cooperative may update less in response to a surprising defection, effectively showing a trait-based dampening of learning rate.

This asymmetry in belief updating has been observed in prior work (e.g., Siegel et al., 2018, Nature Human Behaviour) and could be captured using a dynamic or belief-weighted learning rate. Models incorporating such mechanisms (e.g., dynamic learning rate models as in Jian Li et al., 2011, Nature Neuroscience) could better account for flexible adjustments in response to surprising behavior, particularly in the social domain.

Second, the developmental interpretation of the observed effects would be strengthened by considering possible non-linear relationships between age and model parameters. For instance, certain cognitive or affective traits relevant to social learning-such as sensitivity to reciprocity or reward updating-may follow non-monotonic trajectories, peaking in late adolescence or early adulthood. Fitting age as a continuous variable, possibly with quadratic or spline terms, may yield more nuanced developmental insights.

Finally, the two age groups compared-adolescents (high school students) and adults (university students)-differ not only in age but also in sociocultural and economic backgrounds. High school students are likely more homogenous in regional background (e.g., Beijing locals), while university students may be drawn from a broader geographic and socioeconomic pool. Additionally, differences in financial independence, family structure (e.g., single-child status), and social network complexity may systematically affect cooperative behavior and valuation of rewards. Although these factors are difficult to control fully, the authors should more explicitly address the extent to which their findings reflect biological development versus social and contextual influences.

Comments on revisions:

The authors have addressed most of my previous comments adequately. I only have a minor question: The models with some variations of RL seem to have very similar AIC. What were the authors' criteria in deciding which model is the "winning" model when several models have similar AIC? Are there ways of integrating models with similar structures into a "model family"? Alternatively, is it possible that different models fit better for different subgroups of participants (e.g., high schoolers vs. college students)?

---

## [Author Response]

The following is the authors’ response to the previous reviews

**Public Reviews:**

**Reviewer #1 (Public review):**
Summary:Wu and colleagues aimed to explain previous findings that adolescents, compared to adults, show reduced cooperation following cooperative behaviour from a partner in several social scenarios. The authors analysed behavioural data from adolescents and adults performing a zero-sum Prisoner's Dilemma task and compared a range of social and non-social reinforcement learning models to identify potential algorithmic differences. Their findings suggest that adolescents' lower cooperation is best explained by a reduced learning rate for cooperative outcomes, rather than differences in prior expectations about the cooperativeness of a partner. The authors situate their results within the broader literature, proposing that adolescents' behaviour reflects a stronger preference for self-interest rather than a deficit in mentalising.Strengths:The work as a whole suggests that, in line with past work, adolescents prioritise value accumulation, and this can be, in part, explained by algorithmic differences in weighted value learning. The authors situate their work very clearly in past literature, and make it obvious the gap they are testing and trying to explain. The work also includes social contexts that move the field beyond non-social value accumulation in adolescents. The authors compare a series of formal approaches that might explain the results and establish generative and modelcomparison procedures to demonstrate the validity of their winning model and individual parameters. The writing was clear, and the presentation of the results was logical and wellstructured.

We thank the reviewer for recognizing the strengths of our work.

Weaknesses:(Q1) I also have some concerns about the methods used to fit and approximate parameters of interest. Namely, the use of maximum likelihood versus hierarchical methods to fit models on an individual level, which may reduce some of the outliers noted in the supplement, and also may improve model identifiability.

We thank the reviewer for this suggestion. Following the comment, we added a hierarchical Bayesian estimation. We built a hierarchical model with both group-level (adolescent group and adult group) and individual-level structures for the best-fitting model. Four Markov chains with 4,000 samples each were run, and the model converged well (see Figure supplement 7)

We then analyzed the posterior parameters for adolescents and adults separately. The results were consistent with those from the MLE analysis (see Figure 2—figure supplement 5). These additional results have been included in the Appendix Analysis section (also see Figure supplement 5 and 7). In addition, we have updated the code and provided the link for reference. We appreciate the reviewer’s suggestion, which improved our analysis.

(Q2) There was also little discussion given the structure of the Prisoner's Dilemma, and the strategy of the game (that defection is always dominant), meaning that the preferences of the adolescents cannot necessarily be distinguished from the incentives of the game, i.e. they may seem less cooperative simply because they want to play the dominant strategy, rather than a lower preferences for cooperation if all else was the same.

We thank the reviewer for this comment and agree that adolescents’ lower cooperation may partly reflect a rational response to the incentive structure of the Prisoner’s Dilemma.

However, our computational modeling explicitly addressed this possibility. Model 4 (inequality aversion) captures decisions that are driven purely by self-interest or aversion to unequal outcomes, including a parameter reflecting disutility from advantageous inequality, which represents self-oriented motives. If participants’ behavior were solely guided by the payoff-dominant strategy, this model should have provided the best fit. However, our model comparison showed that Model 5 (social reward) performed better in both adolescents and adults, suggesting that cooperative behavior is better explained by valuing social outcomes beyond payoff structures.

Besides, if adolescents’ lower cooperation is that they strategically respond to the payoff structure by adopting defection as the more rewarding option. Then, adolescents should show reduced cooperation across all rounds. Instead, adolescents and adults behaved similarly when partners defected, but adolescents cooperated less when partners cooperated and showed little increase in cooperation even after consecutive cooperative responses. This pattern suggests that adolescents’ lower cooperation cannot be explained solely by strategic responses to payoff structures but rather reflects a reduced sensitivity to others’ cooperative behavior or weaker social reciprocity motives. We have expanded our Discussion to acknowledge this important point and to clarify how the behavioral and modeling results address the reviewer’s concern.

“Overall, these findings indicate that adolescents’ lower cooperation is unlikely to be driven solely by strategic considerations, but may instead reflect differences in the valuation of others’ cooperation or reduced motivation to reciprocate. Although defection is the payoffdominant strategy in the Prisoner’s Dilemma, the selective pattern of adolescents’ cooperation and the model comparison results indicate that their reduced cooperation cannot be fully explained by strategic incentives, but rather reflects weaker valuation of social reciprocity.”

Appraisal & Discussion:(Q3) The authors have partially achieved their aims, but I believe the manuscript would benefit from additional methodological clarification, specifically regarding the use of hierarchical model fitting and the inclusion of Bayes Factors, to more robustly support their conclusions. It would also be important to investigate the source of the model confusion observed in two of their models.

We thank the reviewer for this comment. In the revised manuscript, we have clarified the hierarchical Bayesian modeling procedure for the best-fitting model, including the group- and individual-level structure and convergence diagnostics. The hierarchical approach produced results that fully replicated those obtained from the original maximumlikelihood estimation, confirming the robustness of our findings. Please also see the response to Q1.

Regarding the model confusion between the inequality aversion (Model 4) and social reward (Model 5) models in the model recovery analysis, both models’ simulated behaviors were best captured by the baseline model. This pattern arises because neither model includes learning or updating processes. Given that our task involves dynamic, multi-round interactions, models lacking a learning mechanism cannot adequately capture participants’ trial-by-trial adjustments, resulting in similar behavioral patterns that are better explained by the baseline model during model recovery. We have added a clarification of this point to the Results:

“The overlap between Models 4 and 5 likely arises because neither model incorporates a learning mechanism, making them less able to account for trial-by-trial adjustments in this dynamic task.”

(Q4) I am unconvinced by the claim that failures in mentalising have been empirically ruled out, even though I am theoretically inclined to believe that adolescents can mentalise using the same procedures as adults. While reinforcement learning models are useful for identifying biases in learning weights, they do not directly capture formal representations of others' mental states. Greater clarity on this point is needed in the discussion, or a toning down of this language.

We sincerely thank the reviewer for this professional comment. We agree that our prior wording regarding adolescents’ capacity to mentalise was somewhat overgeneralized. Accordingly, we have toned down the language in both the Abstract and the Discussion to better align our statements with what the present study directly tests. Specifically, our revisions focus on adolescents’ and adults’ ability to predict others’ cooperation in social learning. This is consistent with the evidence from our analyses examining adolescents’ and adults’ model-based expectations and self-reported scores on partner cooperativeness (see Figure 4). In the revised Discussion, we state:

“Our results suggest that the lower levels of cooperation observed in adolescents stem from a stronger motive to prioritize self-interest rather than a deficiency in predicting others’ cooperation in social learning”.

(Q5) Additionally, a more detailed discussion of the incentives embedded in the Prisoner's Dilemma task would be valuable. In particular, the authors' interpretation of reduced adolescent cooperativeness might be reconsidered in light of the zero-sum nature of the game, which differs from broader conceptualisations of cooperation in contexts where defection is not structurally incentivised.

We thank the reviewer for this comment and agree that adolescents’ lower cooperation may partly reflect a rational response to the incentive structure of the Prisoner’s Dilemma. However, our behavioral and computational evidence suggests that this pattern cannot be explained solely by strategic responses to payoff structures, but rather reflects a reduced sensitivity to others’ cooperative behavior or weaker social reciprocity motives. We have expanded the Discussion to acknowledge this point and to clarify how both behavioral and modeling results address the reviewer’s concern (see also our response to Q2).

(Q6) Overall, I believe this work has the potential to make a meaningful contribution to the field. Its impact would be strengthened by more rigorous modelling checks and fitting procedures, as well as by framing the findings in terms of the specific game-theoretic context, rather than general cooperation.

We thank the reviewer for the professional comments, which have helped us improve our work.

**Reviewer #2 (Public review):**
Summary:This manuscript investigates age-related differences in cooperative behavior by comparing adolescents and adults in a repeated Prisoner's Dilemma Game (rPDG). The authors find that adolescents exhibit lower levels of cooperation than adults. Specifically, adolescents reciprocate partners' cooperation to a lesser degree than adults do. Through computational modeling, they show that this relatively low cooperation rate is not due to impaired expectations or mentalizing deficits, but rather a diminished intrinsic reward for reciprocity. A social reinforcement learning model with asymmetric learning rate best captured these dynamics, revealing age-related differences in how positive and negative outcomes drive behavioral updates. These findings contribute to understanding the developmental trajectory of cooperation and highlight adolescence as a period marked by heightened sensitivity to immediate rewards at the expense of long-term prosocial gains.Strengths:(1) Rigid model comparison and parameter recovery procedure.(2) Conceptually comprehensive model space.(3) Well-powered samples.

We thank the reviewer for highlighting the strengths of our work.

Weaknesses:(Q1) A key conceptual distinction between learning from non-human agents (e.g., bandit machines) and human partners is that the latter are typically assumed to possess stable behavioral dispositions or moral traits. When a non-human source abruptly shifts behavior (e.g., from 80% to 20% reward), learners may simply update their expectations. In contrast, a sudden behavioral shift by a previously cooperative human partner can prompt higher-order inferences about the partner's trustworthiness or the integrity of the experimental setup (e.g., whether the partner is truly interactive or human). The authors may consider whether their modeling framework captures such higher-order social inferences. Specifically, trait-based models-such as those explored in Hackel et al. (2015, Nature Neuroscience)-suggest that learners form enduring beliefs about others' moral dispositions, which then modulate trial-bytrial learning. A learner who believes their partner is inherently cooperative may update less in response to a surprising defection, effectively showing a trait-based dampening of learning rate.

We thank the reviewer for this thoughtful comment. We agree that social learning from human partners may involve higher-order inferences beyond simple reinforcement learning from non-human sources. To address this, we had previously included such mechanisms in our behavioral modeling. In Model 7 (Social Reward Model with Influence), we tested a higher-order belief-updating process in which participants’ expectations about their partner’s cooperation were shaped not only by the partner’s previous choices but also by the inferred influence of their own past actions on the partner’s subsequent behavior. In other words, participants could adjust their belief about the partner’s cooperation by considering how their partner’s belief about them might change. Model comparison showed that Model 7 did not outperform the best-fitting model, suggesting that incorporating higher-order influence updates added limited explanatory value in this context. As suggested by the reviewer, we have further clarified this point in the revised manuscript.

Regarding trait-based frameworks, we appreciate the reviewer’s reference to Hackel et al. (2015). That study elegantly demonstrated that learners form relatively stable beliefs about others’ social dispositions, such as generosity, especially when the task structure provides explicit cues for trait inference (e.g., resource allocations and giving proportions). By contrast, our study was not designed to isolate trait learning, but rather to capture how participants update their expectations about a partner’s cooperation over repeated interactions. In this sense, cooperativeness in our framework can be viewed as a trait-like latent belief that evolves as evidence accumulates. Thus, while our model does not include a dedicated trait module that directly modulates learning rates, the belief-updating component of our best-fitting model effectively tracks a dynamic, partner-specific cooperativeness, potentially reflecting a prosocial tendency.

(Q2) This asymmetry in belief updating has been observed in prior work (e.g., Siegel et al., 2018, Nature Human Behaviour) and could be captured using a dynamic or belief-weighted learning rate. Models incorporating such mechanisms (e.g., dynamic learning rate models as in Jian Li et al., 2011, Nature Neuroscience) could better account for flexible adjustments in response to surprising behavior, particularly in the social domain.

We thank the reviewer for the suggestion. Following the comment, we implemented an additional model incorporating a dynamic learning rate based on the magnitude of prediction errors. Specifically, we developed Model 9: Social reward model with Pearce–Hall learning algorithm (dynamic learning rate), in which participants’ beliefs about their partner’s cooperation probability are updated using a Rescorla–Wagner rule with a learning rate dynamically modulated by the Pearce–Hall (PH) Error Learning mechanism. In this framework, the learning rate increases following surprising outcomes (larger prediction errors) and decreases as expectations become more stable (see Appendix Analysis section for details).

The results showed that this dynamic learning rate model did not outperform our bestfitting model in either adolescents or adults (see Figure supplement 6). We greatly appreciate the reviewer’s suggestion, which has strengthened the scope of our analysis. We now have added these analyses to the Appendix Analysis section (also Figure Supplement 6) and expanded the Discussion to acknowledge this modeling extension and further discuss its implications.

(Q3) Second, the developmental interpretation of the observed effects would be strengthened by considering possible non-linear relationships between age and model parameters. For instance, certain cognitive or affective traits relevant to social learning-such as sensitivity to reciprocity or reward updating-may follow non-monotonic trajectories, peaking in late adolescence or early adulthood. Fitting age as a continuous variable, possibly with quadratic or spline terms, may yield more nuanced developmental insights.

We thank the reviewer for this professional comment. In addition to the linear analyses, we further conducted exploratory analyses to examine potential non-linear relationships between age and the model parameters. Specifically, we fit LMMs for each of the four parameters as outcomes (*α+*, *α-*, *β*, and *ω*). The fixed effects included age, a quadratic age term, and gender, and the random effects included subject-specific random intercepts and random slopes for age and gender. Model comparison using BIC did not indicate improvement for the quadratic models over the linear models for *α*+ (ΔBIC_quadratic-linear_ = 5.09), *α*^-^(ΔBIC_quadratic-linear_ = 3.04), *β* (ΔBIC_quadratic-linear_ = 3.9), or *ω* (ΔBIC_quadratic-linear_ = 0). Moreover, the quadratic age term was not significant for *α+*, *α−*, or *β* (all *ps* > 0.10). For *ω*, we observed a significant linear age effect (*b* = 1.41, *t* = 2.65, *p* = 0.009) and a significant quadratic age effect (*b* = −0.03, *t* = −2.39, *p* = 0.018; see Author response image 1). This pattern is broadly consistent with the group effect reported in the main text. The shaded area in the figure represents the 95% confidence interval. As shown, the interval widens at older ages (≥ 26 years) due to fewer participants in that range, which limits the robustness of the inferred quadratic effect. In consideration of the limited precision at older ages and the lack of BIC improvement, we did not emphasize the quadratic effect in the revised manuscript and present these results here as exploratory.

**Author response image 1. sa3fig1:** Linear and quadratic model fits showing the relationship between age and the *ω* parameter, with 95% confidence intervals.

(Q4) Finally, the two age groups compared - adolescents (high school students) and adults (university students) - differ not only in age but also in sociocultural and economic backgrounds. High school students are likely more homogenous in regional background (e.g., Beijing locals), while university students may be drawn from a broader geographic and socioeconomic pool. Additionally, differences in financial independence, family structure (e.g., single-child status), and social network complexity may systematically affect cooperative behavior and valuation of rewards. Although these factors are difficult to control fully, the authors should more explicitly address the extent to which their findings reflect biological development versus social and contextual influences.

We appreciate this comment. Indeed, adolescents (high school students) and adults (university students) differ not only in age but also in sociocultural and socioeconomic backgrounds. In our study, all participants were recruited from Beijing and surrounding regions, which helps minimize large regional and cultural variability. Moreover, we accounted for individual-level random effects and included participants’ social value orientation (SVO) as an individual difference measure.

Nonetheless, we acknowledge that other contextual factors, such as differences in financial independence, socioeconomic status, and social experience—may also contribute to group differences in cooperative behavior and reward valuation. Although our results are broadly consistent with developmental theories of reward sensitivity and social decisionmaking, sociocultural influences cannot be entirely ruled out. Future work with more demographically matched samples or with socioeconomic and regional variables explicitly controlled will help clarify the relative contributions of biological and contextual factors. Accordingly, we have revised the Discussion to include the following statement:

“Third, although both age groups were recruited from Beijing and nearby regions, minimizing major regional and cultural variation, adolescents and adults may still differ in socioeconomic status, financial independence, and social experience. Such contextual differences could interact with developmental processes in shaping cooperative behavior and reward valuation. Future research with demographically matched samples or explicit measures of socioeconomic background will help disentangle biological from sociocultural influences.”

**Reviewer #3 (Public review):**
Summary:Wu and colleagues find that in a repeated Prisoner's Dilemma, adolescents, compared to adults, are less likely to increase their cooperation behavior in response to repeated cooperation from a simulated partner. In contrast, after repeated defection by the partner, both age groups show comparable behavior.To uncover the mechanisms underlying these patterns, the authors compare eight different models. They report that a social reward learning model, which includes separate learning rates for positive and negative prediction errors, best fits the behavior of both groups. Key parameters in this winning model vary with age: notably, the intrinsic value of cooperating is lower in adolescents. Adults and adolescents also differ in learning rates for positive and negative prediction errors, as well as in the inverse temperature parameter.Strengths:The modeling results are compelling in their ability to distinguish between learned expectations and the intrinsic value of cooperation. The authors skillfully compare relevant models to demonstrate which mechanisms drive cooperation behavior in the two age groups.

We thank the reviewer’s recognition of our work’s strengths.

Weaknesses:(Q1) Some of the claims made are not fully supported by the data:The central parameter reflecting preference for cooperation is positive in both groups. Thus, framing the results as self-interest versus other-interest may be misleading.

We thank the reviewer for this insightful comment. In the social reward model, the cooperation preference parameter is positive by definition, as defection in the repeated rPDG always yields a +2 monetary advantage regardless of the partner’s action. This positive value represents the additional subjective reward assigned to mutual cooperation (e.g., reciprocity value) that counterbalances the monetary gain from defection. Although the estimated social reward parameter *ω* was positive, the effective advantage of cooperation is Δ=*p×ω*−2. Given participants’ inferred beliefs *p*, Δ was negative for most trials (*p×ω*<2), indicating that the social reward was insufficient to offset the +2 advantage of defection. Thus, both adolescents and adults valued cooperation positively, but adolescents’ smaller *ω* and weaker responsiveness to sustained partner cooperation suggest a stronger weighting on immediate monetary payoffs.

In this light, our framing of adolescents as more self-interested derives from their behavioral pattern: even when they recognized sustained partner cooperation and held high expectations of partner cooperation, adolescents showed lower cooperative behavior and reciprocity rewards compared with adults. Whereas adults increased cooperation after two or three consecutive partner cooperations, this pattern was absent among adolescents. We therefore interpret their behavior as relatively more self-interested, reflecting reduced sensitivity to the social reward from mutual cooperation rather than a categorical shift from self-interest to other-interest, as elaborated in the Discussion.

(Q2) It is unclear why the authors assume adolescents and adults have the same expectations about the partner's cooperation, yet simultaneously demonstrate age-related differences in learning about the partner. To support their claim mechanistically, simulations showing that differences in cooperation preference (i.e., the w parameter), rather than differences in learning, drive behavioral differences would be helpful.

We thank the reviewer for raising this important point. In our model, both adolescents and adults updated their beliefs about partner cooperation using an asymmetric reinforcement learning (RL) rule. Although adolescents exhibited a higher positive and a lower negative learning rate than adults, the two groups did not differ significantly in their overall updating of partner cooperation probability (Fig. 4a-b). We then examined the social reward parameter *ω*, which was significantly smaller in adolescents and determined the intrinsic value of mutual cooperation (i.e., *p*×*ω*). This variable differed significantly between groups and closely matched the behavioral pattern.

Following the reviewer’s suggestion, we conducted additional simulations varying one model parameter at a time while holding the others constant. The difference in mean cooperation probability between adults and adolescents served as the index (positive = higher cooperation in adults). As shown in the Author response image 2, decreases in ω most effectively reproduced the observed group difference (shaded area), indicating that age-related differences in cooperation are primarily driven by variation in the social reward parameter *ω* rather than by others.

**Author response image 2. sa3fig2:** Simulation results showing how variations in each model parameter affect the group difference in mean cooperation probability (Adults – Adolescents). Based on the bestfitting Model 8 and parameters estimated from all participants, each line represents one parameter (i.e., *α*+, *α*-, *ω*, *β*) systematically varied within the tested range (*α*±:0.1–0.9; *ω*, *β*:1–9) while other parameters were held constant. Positive values indicate higher cooperation in adults. Smaller *ω* values most strongly reproduced the observed group difference, suggesting that reduced social reward weighting primarily drives adolescents’ lower cooperation.

(Q3) Two different schedules of 120 trials were used: one with stable partner behavior and one with behavior changing after 20 trials. While results for order effects are reported, the results for the stable vs. changing phases within each schedule are not. Since learning is influenced by reward structure, it is important to test whether key findings hold across both phases.

We thank the reviewer for this thoughtful and professional comment. In our GLMM and LMM analyses, we focused on trial order rather than explicitly including the stable vs. changing phase factor, due to concerns about multicollinearity. In our design, phases occur in specific temporal segments, which introduces strong collinearity with trial order. In multi-round interactions, order effects also capture variance related to phase transitions.

Nonetheless, to directly address this concern, we conducted additional robustness analyses by adding a phase variable (stable vs. changing) to GLMM1, LMM1, and LMM3 alongside the original covariates. Across these specifications, the key findings were replicated (see GLMM_sup_2 and LMM_sup_4–5; Tables 9-11), and the direction and significance of main effects remained unchanged, indicating that our conclusions are robust to phase differences.

(Q4) The division of participants at the legal threshold of 18 years should be more explicitly justified. The age distribution appears continuous rather than clearly split. Providing rationale and including continuous analyses would clarify how groupings were determined.

We thank the reviewer for this thoughtful comment. We divided participants at the legal threshold of 18 years for both conceptual and practical reasons grounded in prior literature and policy. In many countries and regions, 18 marks the age of legal majority and is widely used as the boundary between adolescence and adulthood in behavioral and clinical research. Empirically, prior studies indicate that psychosocial maturity and executive functions approach adult levels around this age, with key cognitive capacities stabilizing in late adolescence (Icenogle et al., 2019; Tervo-Clemmens et al., 2023). We have clarified this rationale in the Introduction section of the revised manuscript.

“Based on legal criteria for majority and prior empirical work, we adopt 18 years as the boundary between adolescence and adulthood (Icenogle et al., 2019; Tervo-Clemmens et al., 2023).”

We fully agree that the underlying age distribution is continuous rather than sharply divided. To address this, we conducted additional analyses treating age as a continuous predictor (see GLMM_sup_1 and LMM_sup_1–3; Tables S1-S4), which generally replicated the patterns observed with the categorical grouping. Nevertheless, given the limited age range of our sample, the generalizability of these findings to fine-grained developmental differences remains constrained. Therefore, our primary analyses continue to focus on the contrast between adolescents and adults, rather than attempting to model a full developmental trajectory.

(Q5) Claims of null effects (e.g., in the abstract: "adults increased their intrinsic reward for reciprocating... a pattern absent in adolescents") should be supported with appropriate statistics, such as Bayesian regression.

We thank the reviewer for highlighting the importance of rigor when interpreting potential null effects. To address this concern, we conducted Bayes factor analyses of the intrinsic reward for reciprocity and reported the corresponding BF10 for all relevant post hoc comparisons. This approach quantifies the relative evidence for the alternative versus the null hypothesis, thereby providing a more direct assessment of null effects. The analysis procedure is now described in the Methods and Materials section:

“Post hoc comparisons were conducted using Bayes factor analyses with MATLAB’s bayesFactor Toolbox (version v3.0, Krekelberg, 2024), with a Cauchy prior scale σ = 0.707.”

(Q6) Once claims are more closely aligned with the data, the study will offer a valuable contribution to the field, given its use of relevant models and a well-established paradigm.

We are grateful for the reviewer’s generous appraisal and insightful comments.

**Recommendations for the authors:**

**Reviewer #1 (Recommendations for the authors):**
(1) I commend the authors on a well-structured, clear, and interesting piece of work. I have several questions and recommendations that, if addressed, I believe will strengthen the manuscript.

We thank the reviewer for commending the organization of our paper.

(2) Introduction: - Why use a zero-sum (Prisoner's Dilemma; PD) versus a mixed-motive game (e.g. Trust Task) to study cooperation? In a finite set of rounds, the dominant strategy can be to defect in a PD.

We thank the reviewer for this helpful comment. We agree that both the rationale for using the repeated Prisoner’s Dilemma (rPDG) and the limitations of this framework should be clarified. We chose the rPDG to isolate the core motivational conflict between selfinterest and joint welfare, as its symmetric and simultaneous structure avoids the sequential trust and reputation dependencies/accumulation inherent to asymmetric tasks such as the Trust Game (King-Casas et al., 2005; Rilling et al., 2002).

Although a finitely repeated rPDG theoretically favors defection, extensive prior research shows that cooperation can still emerge in long repeated interactions when players rely on learning and reciprocity rather than backward induction (Rilling et al., 2002; Fareri et al., 2015). Our design employed 120 consecutive rounds, allowing participants to update expectations about partner behavior and to establish stable reciprocity patterns over time. We have added the following clarification to the Introduction:

“The rPDG provides a symmetric and simultaneous framework that isolates the motivational conflict between self-interest and joint welfare, avoiding the sequential trust and reputation dynamics characteristic of asymmetric tasks such as the Trust Game (Rilling et al., 2002; King-Casas et al., 2005)”

(3) Methods:Did the participants know how long the PD would go on for?Were the participants informed that the partner was real/simulated?Were the participants informed that the partner was going to be the same for all rounds?

We thank the reviewer for the meticulous review work, which helped us present the experimental design and reporting details more clearly. the following clarifications: I. Participants were not informed of the total number of rounds in the rPDG. This prevented endgame expectations and avoided distraction from counting rounds, which could introduce additional effects. II. Participants were told that their partner was another human participant in the laboratory. However, the partner’s behavior was predetermined by a computer program. This design enabled tighter experimental control and ensured consistent conditions across age groups, supporting valid comparisons. III. Participants were informed that they would interact with the same partner across all rounds, aligning with the essence of a multiround interaction paradigm and stabilizing partner-related expectations. For transparency, we have clarified these points in the Methods and Materials section:

“Participants were told that their partner was another human participant in the laboratory and that they would interact with the same partner across all rounds. However, in reality, the actions of the partner were predetermined by a computer program. This setup allowed for a clear comparison of the behavioral responses between adolescents and adults. Participants were not informed of the total number of rounds in the rPDG.”

(4) The authors mention that an SVO was also recorded to indicate participant prosociality. Where are the results of this? Did this track game play at all? Could cooperativeness be explained broadly as an SVO preference that penetrated into game-play behaviour?

We thank the reviewer for pointing this out. We agree that individual differences in prosociality may shape cooperative behavior, so we conducted additional analyses incorporating SVO. Specifically, we extended GLMM1 and LMM3 by adding the measured SVO as a fixed effect with random slopes, yielding GLMM_sup_3 and LMM_sup_6 (Tables 12–13). The results showed that higher SVO was associated with greater cooperation, whereas its effect on the reward for reciprocity was not significant. Importantly, the primary findings remained unchanged after controlling for SVO. These results indicate that cooperativeness in our task cannot be explained solely by a broad SVO preference, although a more prosocial orientation was associated with greater cooperation. We have reported these analyses and results in the Appendix Analysis section.

(5) Why was AIC chosen rather an BIC to compare model dominance?

Sorry for the lack of clarification. Both the Akaike Information Criterion (AIC, Akaike, 1974) and Bayesian Information Criterion (BIC, Schwarz, 1978) are informationtheoretic criterions for model comparison, neither of which depends on whether the models to be compared are nested to each other or not (Burnham et al., 2002). We have added the following clarification into the Methods.

“We chose to use the AICc as the metric of goodness-of-fit for model comparison for the following statistical reasons. First, BIC is derived based on the assumption that the “true model” must be one of the models in the limited model set one compares (Burnham et al., 2002; Gelman & Shalizi, 2013), which is unrealistic in our case. In contrast, AIC does not rely on this unrealistic “true model” assumption and instead selects out the model that has the highest predictive power in the model set (Gelman et al., 2014). Second, AIC is also more robust than BIC for finite sample size (Vrieze, 2012).”

(6) I believe the model fitting procedure might benefit from hierarchical estimation, rather than maximum likelihood methods. Adolescents in particular seem to show multiple outliers in a^+ and w^+ at the lower end of the distributions in Figure S2. There are several packages to allow hierarchical estimation and model comparison in MATLAB (which I believe is the language used for this analysis; see https://journals.plos.org/ploscompbiol/article?id=10.1371/journal.pcbi.1007043).

We thank the reviewer for this helpful comment and for referring us to relevant methodological work (Piray et al., 2019). We have addressed this point by incorporating hierarchical Bayesian estimation, which effectively mitigates outlier effects and improves model identifiability. The results replicated those obtained with MLE fitting and further revealed group-level differences in key parameters. Please see our detailed response to Reviewer#1 Q1 for the full description of this analysis and results.

(7) Results: Model confusion seems to show that the inequality aversion and social reward models were consistently confused with the baseline model. Is this explained or investigated? I could not find an explanation for this.

The apparent overlap between the inequality aversion (Model 4) and social reward (Model 5) models in the recovery analysis likely arises because neither model includes a learning mechanism, making them unable to capture trial-by-trial adjustments in this dynamic task. Consequently, both were best fit by the baseline model. Please see Response to Reviewer #1 Q3 for related discussion.

(8) Figures 3e and 3f show the correlation between asymmetric learning rates and age. It seems that both a^+ and a^- are around 0.35-0.40 for young adolescents, and this becomes more polarised with age. Could it be that with age comes an increasing discernment of positive and negative outcomes on beliefs, and younger ages compress both positive and negative values together? Given the higher stochasticity in younger ages (\beta), it may also be that these values simply represent higher uncertainty over how to act in any given situation within a social context (assuming the differences in groups are true).

We appreciate this insightful interpretation. Indeed, both α+ and α- cluster around 0.35–0.40 in younger adolescents and become increasingly polarized with age, suggesting that sensitivity to positive versus negative feedback is less differentiated early in development and becomes more distinct over time. This interpretation remains tentative and warrants further validation. Based on this comment, we have revised the Discussion to include this developmental interpretation.

We also clarify that in our model *β* denotes the inverse temperature parameter; higher *β* reflects greater choice precision and value sensitivity, not higher stochasticity. Accordingly, adolescents showed higher *β* values, indicating more value-based and less exploratory choices, whereas adults displayed relatively greater exploratory cooperation. These group differences were also replicated using hierarchical Bayesian estimation (see Response to Reviewer #1 Q1). In response to this comment, we have added a statement in the Discussion highlighting this developmental interpretation.

“Together, these findings suggest that the differentiation between positive and negative learning rates changes with age, reflecting more selective feedback sensitivity in development, while higher β values in adolescents indicate greater value sensitivity. This interpretation remains tentative and requires further validation in future research.”

(9) A parameter partial correlation matrix (off-diagonal) would be helpful to understand the relationship between parameters in both adolescents and adults separately. This may provide a good overview of how the model properties may change with age (e.g. a^+'s relation to \beta).

We thank the reviewer for this helpful comment. We fully agree that a parameter partial correlation matrix can further elucidate the relationships among parameters. Accordingly, we conducted a partial correlation analysis and added the visually presented results to the revised manuscript as Figure 2-figure supplement 4.

(10) It would be helpful to have Bayes Factors reported with each statistical tests given that several p-values fall within the 0.01 and 0.10.

We thank the reviewer for this important recommendation. We have conducted Bayes factor analyses and reported BF10 for all relevant post hoc comparisons. We also clarified our analysis in the Methods and Materials section:

“Post hoc comparisons were conducted using Bayes factor analyses with MATLAB’s bayesFactor Toolbox (version v3.0, Krekelberg, 2024), with a Cauchy prior scale σ = 0.707.”

(11) Discussion: I believe the language around ruling out failures in mentalising needs to be toned down. RL models do not enable formal representational differences required to assess mentalising, but they can distinguish biases in value learning, which in itself is interesting. If the authors were to show that more complex 'ToM-like' Bayesian models were beaten by RL models across the board, and this did not differ across adults and adolescents, there would be a stronger case to make this claim. I think the authors either need to include Bayesian models in their comparison, or tone down their language on this point, and/or suggest ways in which this point might be more thoroughly investigated (e.g., using structured models on the same task and running comparisons: https://journals.plos.org/plosone/article?id=10.1371/journal.pone.0087619).

We thank the reviewer for the comments. Please see our response to Reviewer 1 (Appraisal & Discussion section) for details.

**Reviewer #2 (Recommendations for the authors):**
(1) The authors may want to show the winning model earlier (perhaps near the beginning of the Results section, when model parameters are first mentioned).

We thank the reviewer for this suggestion. We agree that highlighting the winning model early improves clarity. Currently, we have mentioned the winning model before the beginning of the Results section. Specifically, in the penultimate paragraph of the Introduction we state:

“We identified the asymmetric RL learning model as the winning model that best explained the cooperative decisions of both adolescents and adults.”

**Reviewer #3 (Recommendations for the authors):**
(1) In addition to the points mentioned above, I suggest the following:Clarify plots by clearly explaining each variable. In particular, the indices 1 vs. 1,2 vs 1,2,3 were not immediately understandable.

We thank the reviewer for this suggestion. We agree that the indices were not immediately clear. We have revised the figure captions (Figure 1 and 4) to explicitly define these terms more clearly:

“The x-axis represents the consistency of the partner’s actions in previous trials (t_−1_: last trial; t_−1,2_: last two trials;_t−1,2,3_: last three trials).”

(2) It's unclear why the index stops at 3. If this isn't the maximum possible number of consecutive cooperation trials, please consider including all relevant data, as adolescents might show a trend similar to adults over more trials.

We thank the reviewer for raising this point. In our exploratory analyses, we also examined longer streaks of consecutive partner cooperation or defection (up to four or five trials). Two empirical considerations led us to set the cutoff at three in the final analyses. First, the influence of partner behavior diminished sharply with temporal distance. In both GLMMs and LMMs, coefficients for earlier partner choices were small and unstable, and their inclusion substantially increased model complexity and multicollinearity. This recency pattern is consistent with learning and decision models emphasizing stronger weighting of recent evidence (Fudenberg & Levine, 2014; Fudenberg & Peysakhovich, 2016). Second, streaks longer than three were rare, especially among some participants, leading to data sparsity and inflated uncertainty. Including these sparse conditions risked biasing group estimates rather than clarifying them. Balancing informativeness and stability, we therefore restricted the index to three consecutive partner choices in the main analyses, which we believe sufficiently capture individuals’ general tendencies in reciprocal cooperation.

(3) The term "reciprocity" may not be necessary. Since it appears to reflect a general preference for cooperation, it may be clearer to refer to the specific behavior or parameter being measured. This would also avoid confusion, especially since adolescents do show negative reciprocity in response to repeated defection.

We thank you for this comment. In our work, we compute the intrinsic reward for reciprocity as p × ω, where p is the partner cooperation expectation and ω is the cooperation preference. In the rPDG, this value framework manifests as a reciprocity-derived reward: sustained mutual cooperation maximizes joint benefits, and the resulting choice pattern reflects a value for reciprocity, contingent on the expected cooperation of the partner. This quantity enters the trade-off between U_cooperation_ and U_defection_ and captures the participant’s intrinsic reward for reciprocity versus the additional monetary reward payoff of defection. Therefore, we consider the term “reciprocity” an acceptable statement for this construct.

(4) Interpretation of parameters should closely reflect what they specifically measure.

We thank the reviewer for pointing this out. We have refined the relevant interpretations of parameters in the current Results and Discussion sections.

(5) Prior research has shown links between Theory of Mind (ToM) and cooperation (e.g., Martínez-Velázquez et al., 2024). It would be valuable to test whether this also holds in your dataset.

We thank the reviewer for this thoughtful comment. Although we did not directly measure participants’ ToM, our design allowed us to estimate participants’ trial-by-trial inferences (i.e., expectations) about their partner’s cooperation probability. We therefore treat these cooperation expectations as an indirect representation for belief inference, which is related to ToM processes. To test whether this belief-inference component relates to cooperation in our dataset, we further conducted an exploratory analysis (GLMM_sup_4) in which participants’ choices were regressed on their cooperation expectations, group, and the group × cooperation-expectation interaction, controlling for trial number and gender, with random effects. Consistent with the ToM–cooperation link in prior research (MartínezVelázquez et al., 2024), participants’ expectations about their partner’s cooperation significantly predicted their cooperative behavior (Table 14), suggesting that decisions were shaped by social learning about others’ inferred actions. Moreover, the interaction between group and cooperation expectation was not significant, indicating that this inference-driven social learning process likely operates similarly in adolescents and adults. This aligns with our primary modeling results showing that both age groups update beliefs via an asymmetric learning process. We have reported these analyses in the Appendix Analysis section.

(6) More informative table captions would help the reader. Please clarify how variables are coded (e.g., is female = 0 or 1? Is adolescent = 0 or 1?), to avoid the need to search across the manuscript for this information.

We thank the reviewer for raising this point. We have added clear and standardized variable coding in the table notes of all tables to make them more informative and avoid the need to search the paper. We have ensured consistent wording and formatting across all tables.

(7) I hope these comments are helpful and support the authors in further strengthening their manuscript.

We thank the three reviewers for their comments, which have been helpful in strengthening this work.

References

(1) Fudenberg, D., & Levine, D. K. (2014). Recency, consistent learning, and Nash equilibrium. Proceedings of the National Academy of Sciences of the United States of America, 111(Suppl. 3), 10826–10829. https://doi.org/10.1073/pnas.1400987111.

(2) Fudenberg, D., & Peysakhovich, A. (2016). Recency, records, and recaps: Learning and nonequilibrium behavior in a simple decision problem. ACM Transactions on Economics and Computation, 4(4), Article 23, 1–18. https://doi.org/10.1145/2956581

(3) Hackel, L., Doll, B., & Amodio, D. (2015). Instrumental learning of traits versus rewards: Dissociable neural correlates and effects on choice. Nature Neuroscience, 18, 1233– 1235. https://doi.org/10.1038/nn.4080

(4) Icenogle, G., Steinberg, L., Duell, N., Chein, J., Chang, L., Chaudhary, N., Di Giunta, L., Dodge, K. A., Fanti, K. A., Lansford, J. E., Oburu, P., Pastorelli, C., Skinner, A. T.Sorbring, E., Tapanya, S., Uribe Tirado, L. M., Alampay, L. P., Al-Hassan, S. M.,Takash, H. M. S., & Bacchini, D. (2019). Adolescents’ cognitive capacity reaches adult levels prior to their psychosocial maturity: Evidence for a “maturity gap” in a multinational, cross-sectional sample. Law and Human Behavior, 43(1), 69–85. https://doi.org/10.1037/lhb0000315

(5) Krekelberg, B. (2024). Matlab Toolbox for Bayes Factor Analysis (v3.0) [Computer software]. Zenodo. https://doi.org/10.5281/zenodo.13744717

(6) Martínez-Velázquez, E. S., Ponce-Juárez, S. P., Díaz Furlong, A., & Sequeira, H. (2024). Cooperative behavior in adolescents: A contribution of empathy and emotional regulation? Frontiers in Psychology, 15,1342458. https://doi.org/10.3389/fpsyg.2024.1342458

(7) Tervo-Clemmens, B., Calabro, F. J., Parr, A. C., et al. (2023). A canonical trajectory of executive function maturation from adolescence to adulthood. Nature Communications, 14, 6922. https://doi.org/10.1038/s41467-023-42540-8

(8) King-Casas, B., Tomlin, D., Anen, C., Camerer, C. F., Quartz, S. R., & Montague, P. R. (2005). Getting to know you: reputation and trust in a two-person economic exchange. Science, 308(5718), 78-83. https://doi.org/10.1126/science.1108062

(9) Rilling, J. K., Gutman, D. A., Zeh, T. R., Pagnoni, G., Berns, G. S., & Kilts, C. D. (2002).A neural basis for social cooperation. Neuron, 35(2), 395-405. https://doi.org/10.1016/s0896-6273(02)00755-9

(10) Fareri, D. S., Chang, L. J., & Delgado, M. R. (2015). Computational substrates of social value in interpersonal collaboration. Journal of Neuroscience, 35(21), 8170-8180. https://doi.org/10.1523/JNEUROSCI.4775-14.2015

(11) Akaike, H. (2003). A new look at the statistical model identification. IEEE transactions on automatic control, 19(6), 716-723. https://doi.org/10.1109/TAC.1974.1100705

(12) Schwarz, G. (1978). Estimating the dimension of a model. The annals of statistics, 461464. https://doi.org/10.1214/aos/1176344136

(13) Burnham, K. P., & Anderson, D. R. (2002). Model selection and multimodel inference: A practical information-theoretic approach (2nd ed.). Springer.https://doi.org/10.1007/b97636

(14) Gelman, A., & Shalizi, C. R. (2013). Philosophy and the practice of Bayesian statistics. British Journal of Mathematical and Statistical Psychology, 66(1), 8–38. https://doi.org/10.1111/j.2044-8317.2011.02037.x

(15) Gelman, A., Carlin, J. B., Stern, H. S., Dunson, D. B., Vehtari, A., & Rubin, D. B. (2014). Bayesian data analysis (3rd ed.). Chapman and Hall/CRC. https://doi.org/10.1201/b16018

(16) Vrieze, S. I. (2012). Model selection and psychological theory: A discussion of the differences between the Akaike Information Criterion (AIC) and the Bayesian Information Criterion (BIC). Psychological Methods, 17(2), 228–243. https://doi.org/10.1037/a0027127